# New Insight to Niche Partitioning and Ecological Function of Ammonia Oxidizing Archaea in Subtropical Estuarine Ecosystem

Yanhong Lu[1,2,†],Shunyan Cheung[2,†], Ling Chen[3], Shuh-Ji Kao[3], Xiaomin Xia[4], Jianping Gan[1], Minhan Dai[3], Hongbin Liu[2,5]

[1]SZU-HKUST Joint PhD Program in Marine Environmental Science, Shenzhen University, Shenzhen, China
[2]Department of Ocean Science, The Hong Kong University of Science and Technology, Hong Kong, China
[3]State Key Laboratory of Marine Environmental Science, Xiamen University, China
[4]Key Laboratory of Tropical Marine Bio-resources and Ecology, South China Sea Institute of Oceanology, Chinese Academy of Sciences, Guangzhou, China
[5]Hong Kong Branch of Southern Marine Science & Engineering Guangdong Laboratory, The Hong Kong University of Science and Technology, Hong Kong, China
†These authors shared equal contributions

*Correspondence to*: Hongbin Liu (liuhb@ust.hk)

**Abstract.** Nitrification plays a central role in estuarine nitrogen cycle. Previous studies in estuary mainly focused on the niche-partition between ammonia-oxidizing archaea (AOA) and bacteria (AOB), while the diversity, activity, biogeography and ecophysiology of different AOA groups remained unclear. Here, we for the first time reported niche partitioning as well as differentially distributed active populations among diverse AOA (inferred from *amoA* gene) in a typical subtropical estuary–Pearl River estuary (PRE). In the water column of PRE, the AOA communities mainly consisted of WCA and SCM1-like sublineages. Surprisingly, we observed a strong disagreement of AOA communities at DNA and RNA levels. In DNA samples, WCA generally dominated the AOA community, and the distributional pattern indicated that WCA I and WCA II sublineages preferred oceanic and coastal conditions, respectively. In contrast, diverse SCM1-like sublineages were identified and outnumbering WCA at RNA level, in which SCM1-like-III was limited to freshwater while the rest sublineages were widely distributed in the estuary. The SCM1-like sublineages strongly correlated with nitrification rate, which indicated their important contribution to ammonia oxidation. Furthermore, intense nitrification contributed significantly to hypoxia conditions (nitrification contributed averaged 12.18 % of oxygen consumption) in the estuary. These results revealed different ammonia-oxidizing activities and niche partitioning among different AOA sublineages in estuarine water, which was unexplored in previous DNA and clone library-based studies. The ecological significance and functioning of the diverse AOA should be further explored in the marine ecosystem.

# 1 Introduction

Nitrification is a microbial mediated oxidation process of ammonia to nitrate, interconnects the source (N-fixation), and sink (N-loss) and plays a central role in the marine nitrogen cycling (Ward 1996). Particularly in the estuarine ecosystem, nitrification significantly impacts the N source for primary production and oxygen level in the water column (Yool et al. 2007; Erguder et al. 2009; Campbell et al. 2019). Regarding to the biogeochemical significance of ammonia oxidation (i.e. the first and rate-determining step of nitrification) in the estuarine ecosystem, the physiology and ecological function of ammonia oxidizers (i.e. ammonia-oxidizing archaea (AOA) and bacteria (AOB)) have been the major interest to understand the estuarine N transformation (Bernhard and Bollmann 2010). Previous studies were mostly conducted in the sediment of estuarine ecosystems (summarized in Table S1) (Damashek et al. 2016). These studies mainly focused on the niche partition between AOA and AOB inferred from *amoA* genes abundance and collectively showed the AOA outnumbered AOB in the estuarine ecosystem (Caffrey et al. 2007; Abell et al. 2010; Bernhard et al. 2010). However, the biogeography, niche partition, and ecological function of different AOA groups were little analyzed (Table S1).

Based on the *amoA* gene (ammonia monooxygenase subunit A), the marine AOA was recognized to consist of three major groups: water column A (WCA; shallow water ecotype dominating in epipelagic and upper mesopelagic water), water column B (WCB; deep water ecotype dominating in mesopelagic and bathypelagic water) and SCM1-like (affiliated to the first isolated AOA–*Nitrosopumilus maritimus* SCM1), corresponding to the group NP-Epsilon, NP-Alpha and NP-Gamma, respectively, in the global synthesis of Alves et al. 2018 (Alves 2018; Cheung et al. 2019). The distribution and abundance of WCA and WCB were much more studied than SCM1-like ecotype in the field observations (Francis et al. 2005; Hallam et al. 2006; Beman et al. 2008; Beman et al. 2012). Recently, highly diverse sublineages of WCA and WCB were revealed in the global ocean, in which sublineage within each ecotype displayed varied distributional patterns and environmental determinants (Cheung et al. 2019). Since most of marine AOA remained uncultivated, our understanding of the ecophysiology of most of AOA (especially WCA and WCB) in marine ecosystems relied heavily on field observations (Alves et al. 2018). For example, niche partitioning between WCB sublineages has been recently observed in the oxygen minimum zone off the Costa Rica Dome and potential anoxic adapted phylotypes were widely detected between the geographically distant OMZs (Lu et al. 2019).

As mentioned, population dynamics and ecological function of different AOA were rarely studied in the estuarine water comparing to the relatively well-characterized AOA populations in oceanic waters, as well as sediment and soil environments (Bernhard and Bollmann 2010; Damashek et al. 2016). Previous studies of marine AOA relied mainly on clone library analysis (summarized in Table S1), which were insufficient to recover the diversity and biogeography of AOA. Moreover, studies relied on DNA surveys do not provide information of active AOA communities. Recently, Wu et al. reported differentially transcriptional activities of terrestrial AOA communities referred from DNA and RNA extracts, suggesting that studies using DNA may have underestimated the importance of some active AOA groups in the natural environments (Wu et al. 2017). In this study, we have conducted a comprehensive study about ammonia oxidizers in a typical subtropical estuary-Pearl River estuary (PRE), characterized by its salt-wedge structure resulted from large amount of freshwater discharge during wet season

(Harrison et al. 2008). Recently, the recurrence of bottom water hypoxia at the lower estuary of PRE has received increasing
concerns about its ecological impact on the estuarine ecosystem (Qian et al. 2018; Zhao et al. 2020). The steep natural gradients
of salinity, nutrients, oxygen concentration and turbidity makes the Pearl River estuary an ideal environment to study the
diversity and ecological function of ammonia oxidizers. By revealing AOA community structure (dominant ammonia oxidizer)
at DNA and RNA levels using high throuput sequencing and fine-scale phylogenetic classification, along with quantification
of AOA and AOB and nitrification rate measurement, we aim to 1) identify the major and active AOA in the estuarine
ecosystem, 2) reveal niche partitioning between different AOA sublineages based on environmental determinants, and 3)
determine the potential contribution of nitrification to hypoxia formation in PRE.
**2 Materials and methods**
**2.1 Sample collection**
The cruise was conducted from July 11 to August 1 in 2017 on R/V Hai Ke 68. In the first leg, 83 stations were designed
within the 10-50m isobaths covering areas from the upper estuary to the continental shelf (Fig. S1). Water samples were
collected using Niskin bottles equipped with CTD sensor (Sea-Bird SBE 917plus). Temperature, salinity, and depth data were
acquired through the CTD sensor. The dissolved oxygen concentrations were measured on board using Winkler
spectrophotometric and titration method (Pai et al. 2001; Dai et al. 2006; Zhao et al. 2020). Dissolved inorganic nutrient
samples were filtered through pre-acid washed cellulose acetate fiber membranes and stored in -20 ℃ until analysis in a land-
based laboratory in Xiamen University (Qian et al. 2018). Ammonium concentration was measured on board using the
indophenol blue spectrophotometric method (Pai et al. 2001). Chlorophyll-*a* samples (250 to 500ml) were filter onto GF/F
(Whatman, USA) and stored in foil bags in liquid nitrogen. The chlorophyll-*a* concentration was measured with a Turner
Fluorometer (Welschmeyer 1994) after being extracted with 90 % acetone for 14 h at -20 ℃. The microbial abundances were
quantified by a Becton-Dickson FACSCalibur flow cytometer (Vaulot et al. 1989). Seawater for microbial abundance
quantification was prefiltered through a 20 μm mesh, fixed with final concentration of 0.5 % seawater-buffed
paraformaldehyde in cryotubes, and stored in liquid nitrogen until flow cytometric analysis (Liu et al. 2014). At each sampling
depth, 0.5-2 L of seawater were sequentially filtrated onto 3 μm and 0.2 μm polycarbonate membranes (GVS, USA) for
particle-attached and free-living microbes. DNA/RNA samples were immersed in 500 μl RNAlater (Ambion, Austin, TX,
USA) before stored in liquid nitrogen.
**2.2 Rates measurement**
Community respiration rates (CR) were estimated by measuring the oxygen consumption in triplicate 60ml BOD bottles
without headspace after 24 h dark incubation submerged in seawater continuously pumped from sea surface. Nitrification were
measured by incubating $^{15}NH_4^+$ amended (less than 10 % of ambient concentration) seawater in duplicate 200 ml HDPE bottles

in dark for 6-12 h, with temperature controlled by running seawater. After incubation, filtrate (0.2 μm-syringe-filtered) was collected and stored in -20 °C for downstream $^{15}NO_x^-$ ($^{15}NO_3^- + {}^{15}NO_2^-$) analysis (Sigman et al. 2001).

The nitrification rates were calculated using the following equation:

$$AO_b = \frac{(R_t NO_x^- \times [NO_x^-]_t) - (R_{t0} NO_x^- \times [NO_x^-]_{t0})}{t-t0} \times \frac{\left[^{14}NH_4^+\right] + \left[^{15}NH_4^+\right]}{\left[^{15}NH_4^+\right]} \qquad (1)$$

In equation 1, $AO_b$ is the bulk nitrification rate. $R_{t0} NO_x^-$ and $R_t NO_x^-$ are the ratios (%) of $^{15}N$ in the $NO_x^-$ pool measured at the initial ($t_0$) and termination (t) of the incubation. $[NO_x^-]_{t0}$ and $[NO_x^-]_t$ are the concentration of $NO_x^-$ at the initial and termination of the incubation, respectively. $[^{14}NH_4^+]$ is the ambient $NH_4^+$ concentration. $[^{15}NH_4^+]$ is the final ammonium concentration after addition of the stable isotope tracer ($^{15}NH_4^+$). The $NO_x^-$ was completely converted to $N_2O$ by a single strain of denitrifying bacteria (*Pseudomonas aureofaciens,* ATCC#13985) which lack $N_2O$-reductase activity (Sigman et al. 2001). The converted $N_2O$ was further analyzed using IRMS (Isotope Ration Mass Spectrometer, Thermo Scientific Delta V Plus) to calculate the isotopic composition of $NO_x^-$ (Sigman et al. 2001; Casciotti et al. 2002; Knapp et al. 2005).We analyzed the correlation between nitrification rates and AOA sublineages. Equation 2 was generally considered as the oxidation of ammonia to nitrite. Inferred from the nitrification rates, we estimated the nitrification oxygen demand (NOD) based on equations 2. Inferred from the nitrification rates, we estimated the NOD based on equation 2. We used NOD/CR ratio (percentage) to evaluate the potential contribution of nitrification to total oxygen consumption in the field.

$$NH_3 + 1.5O_2 \rightarrow NO_2^- + H_2O + H^+ \qquad (2)$$

## 2.3 DNA and RNA extraction and cDNA synthesis

The sample filters immersed in RNAlater were thawed on ice. RNAlater was removed following the procedure described in Xu et al. 2013 (Xu et al. 2013). For DNA extraction, filters were cut into pieces and carefully collected into the 2ml Lysing Matrix E tubes with the addition of 978 μl sodium phosphate buffer and 122 μl MT buffer provided in FastDNA™ SPIN Kit for Soil (MP Biomedical, Solon, OH, USA). The lysing matrix was homogenized by Mini-Beadbeater-24 (Biospec Product, Bartlesville, OK, USA), at 3500 oscl/min for 60 seconds. The subsequent procedures of DNA extraction were performed according to FastDNA Spin kit for soil manufacture's instruction and preserved at -80 °C. For RNA extraction, sample filters were incubated in 1 ml TRIzol for 5 min at room temperature in 2ml sterile microcentrifuge tubes. After the incubation, 200 μl chloroform was added into the tubes and mixed vigorously by hand until the membrane fully dissolved. After room temperature incubation for 3 min, the samples were centrifuged at $12000 \times g$ and 4 °C for 15 min. The supernatant was carefully transferred into a new 2ml microcentrifuge and mixed with an equal volume of 70 % ethanol. The purification and elution procedures were performed according to the manufacture's instruction of the PureLink RNA Mini Kit (Life Technologies, Carlsbad, CA, USA). RNA samples were immediately treated with DNase at 37 °C for 30 min using the TURBO DNA-free Kit to eliminate DNA contamination. After incubation, the DNase was inactivated following the manufacturer's instruction.

The DNA-free RNA samples were reversely transcribed into cDNA with random primers using the SuperScript III First-Strand
Synthesis System (Life Technologies, Carlsbad, CA, USA). The synthesized cDNA was further treated with RNase H at 37 ℃
for 20 min to remove the residual RNA.
**2.4 PCR amplification and high throughput sequencing**
The DNA and cDNA were used as templates in PCR amplification. The archaeal *amoA* gene fragments were amplified using
the barcoded primers Arch-amoAF (5'-adaptor+barcode+GAT+STAATGGTCTGGCTTAGACG-3') and Arch-amoAR (5'-
adaptor+barcode+GAT+GCGGCCATCCATCTGTATGT-3') (Francis et al. 2005). Triplicated PCR reactions were performed
in 12.5 μl mixture contained 1×PCR buffer, 2 mM MgCl$_2$, 0.2 mM dNTP mix, 0.4 μM of respective primers, 2 U Invitrogen
Platinum Taq DNA polymerase (Life Technologies, Carlsbad, CA, USA) and 1 μl template. The PCR thermal cycle consisted
of 5 min initial denaturation at 95 ℃ and followed by 33 cycles of 95 ℃ for 30s, 53 ℃ for 45s, and 72 ℃ for 60s and 10 min
of final extension step at 72 ℃. The triplicated PCR products of each sample were pooled together and sequenced on the Ion
GeneStudio S5 system (Thermo Fisher Scientific, USA) which could generate around 600 bp high quality reads.
**2.5 Standard curve construction and Quantitative PCR**
The *amoA* gene of AOA and β-AOB *amoA* was amplified by the primer pair Arch-amoAF-amoAR (Francis et al. 2005) and
amoA-1F and amo-2R (Rotthauwe et al. 1997) respectively, using the DNA mixture from A-transect samples. The PCR
products were purified using the illustra GFX PCR DNA and Gel band purification kit (GE Healthcare, UK) and ligated into
T-vector pMD 19 at 4 ℃ for 12 h (Takara, Japan). The ligated vectors solution was mixed with freshly prepared *E. coli* BL21
competent cell and incubated on ice for 30 min. Heat-shock treatment at 42 ℃ were performed for the mixture for 90 s and
incubated on ice for 5 min. After 5min incubation, 200 μl of liquid lysogeny broth was added and incubated at 37 ℃ for 1h in
incubator shaker (250 rpm/min). The culture was soon spread on to ampicillin (100 mg·L$^{-1}$) containing plates and incubated at
37 ℃ for 12 h. White clone was selected and confirmed with respective PCR amplification. The clones were expanded with
ampicillin (100 mg·L$^{-1}$) lysogeny broth and sequenced in BGI Tech (BGI, Shenzhen, China). The sequence of the selected
plasmid was confirmed as an archaeal *amoA* gene by blast against the NCBI database. The plasmid of the selected clone was
extracted and purified by the TIANprep Mini Plasmid Kit (TIANGEN, China). The extracted plasmid was linearized by EcoRI
(New England Biolabs) at 37 ℃ for 12 h and purified by electrophoresis on 1.2 % agarose gel. The linearized plasmid DNA
concentration was determined via dsDNA HS assay on the Qubit fluorometer v3.0 (Thermo Fisher Scientific, Singapore).
Series dilution of the linearized plasmids was amplified as standard curves together with the field samples on the 384-well
plates on Roche LightCycler 480.
Triplicated quantitative PCR (qPCR) was performed in 10 μl mixture contained 1 × LightCycler® 480 SYBR® Green I Master,
0.5 μM primers pairs and DNA templates. The thermal cycle of the qPCR that targeted archaeal *amoA* gene consisted of a 5
min denaturation at 95 ℃, followed by 45 cycles each at 95 ℃ for 30s, 53 ℃ (60 ℃ for β-AOB) for 45s, 72 ℃ for 60s with
single signal acquisition at the end of each cycle. Amplification specificity was confirmed via the melting curve and gel
electrophoresis. In both particle-attached (> 3 μm) and free-living (0.2-3 μm) DNA (and RNA), the AOA and β-AOB were
quantified based on the *amoA* gene abundance through the qPCR (Table S2).

## 2.6 Bioinformatic analysis

The archaeal *amoA* gene sequencing data of 76 samples (contained 2523 reads per sample) were analyzed using the microbial
ecology community software program Mothur (Schloss et al. 2009). The sequencing output was split according to
corresponding barcode sequences in the forward primer. Quality control was performed by discarding the reads with low-
quality (average quality score < 20), incorrect length (no shorter than 300 bp and no longer than 630 bp), ambiguous base or
homopolymers longer than 8 bp. The chimeric sequences were identified and discarded by the *Chimera.uchime* in Mothur.
The remaining high-quality sequences were aligned with the reference *amoA* sequences from the NCBI database using Mothur
(Agarwala et al. 2018) and were clustered into operational taxonomic units (OTUs) at 95 % DNA similarity. The singletons
and doubletons were discarded from the OTU table before downstream analyses. The representative sequences of the top OTUs
were randomly selected through *getotu.rep* in Mothur and searched against the NCBI database using Blastn. The top OTUs
were selected based on relative abundance ≥ 0.1 % (Logares et al. 2014). The Maximum Likelihood phylogenetic tree was
constructed in MEGA 7 with the recommended model (T92+G+I) after the best model selection. The ML-tree was further
edited with iTOL (Letunic and Bork 2016). The Bray-Curtis dissimilarities among the AOA communities were calculated with
"*vegdist*" function of the "*vegan*" package in R. Nonmetric multidimensional scaling (NMDS) analysis was performed based
on the Bray-Curtis dissimilarities with the "*vegan*" package and visualized with "*ggplot2*" package in R (Oksanen, et al. 2019;
Wickham, 2016).
Considering the strong stratification and steep variation of environmental factors that associated with the freshwater
discharge in the PRE, Spearman correlation analysis was performed to determine the relationship between the AOA
sublineages and environmental factors in surface DNA, surface RNA, bottom DNA and bottom RNA samples, respectively.
Besides, Spearman correlation analysis was performed between nitrification rates and *amoA* gene (AOA and β-AOB)
abundances retrieved from particle-attached (> 3 μm) and free-living (3-0.2 μm) samples.

## 3 Results

### 3.1 Hydrographic characteristics of Pearl River estuary

The Pearl River estuary consists of three major sub-estuaries, namely Lingdingyang, Modaomen, and Huangmaohai (Fig. 1),
which contribute to 55 %, 28 %, and 13 % of the annual mean of freshwater discharge, respectively (Zhao 1990). This
investigation was conducted in the wet season when high freshwater discharge formed a large plume extending southwestward
(Fig. 2a and d). Associated with the plume, an excessive phytoplankton bloom was observed in the lower estuary with
chlorophyll-*a* concentration peaked (28.4 μg·L$^{-1}$) at station F202 (Fig. 2b and e). Furthermore, widespread bottom water
hypoxia (DO < 2 mg·L$^{-1}$) was observed in the lower reach of Pearl River estuary extending from Huangmaohai to the southern
water of Hong Kong island (Fig. 2f). Our study area covered a full range of salinity from 0.1 to 34.7. The variation of nitrate
concentration followed salinity gradient (Fig. S3a and d). High concentrations of nitrate were detected in low salinity waters
near the outlets of sub-estuaries, with the highest value ($> 115$ $\mu mol \cdot L^{-1}$) observed in the surface water of Lingdingyang (station
A01-03). Similar to nitrate, the concentrations of nitrite in the surface layer were also high near the estuary outlets and peaked
at station A01 (9.5 $\mu mol \cdot L^{-1}$), but relatively constant ($< 2$ $\mu mol \cdot L^{-1}$) in the bottom layer (Fig. S3c and f). The ammonium
concentration displayed a different spatial pattern compared to nitrate and nitrite, with maximum concentration occurred at
A06 (2.5 $\mu mol \cdot L^{-1}$ and 3.2 $\mu mol \cdot L^{-1}$ in surface and bottom layer, respectively) possibly influenced by local sewage discharges.
A patch of relatively high ammonium water ($> 1$ $\mu mol \cdot L^{-1}$) was observed in the southern water of Hong Kong, spreading
eastward at the stations along the south borderline of Hong Kong water (Fig. S3c).

## 3.2 The spatial pattern of nitrification rates and their oxygen consumption

The nitrification rates were generally higher in bottom water than in surface water, except station A01 and F601 (Fig. 3). At
the surface layer, high nitrification rates were detected in the outlet of Humen and Modaomen (station A01 and F301) and the
southern water of Hong Kong (station F601 and F701) (Table S2). At the bottom layer, high nitrification rates were detected
in the Humen outlet and the lower estuary from Huangmaohai to the southern water of Hong Kong (Fig. 3a). Based on equation
2, the NOD were estimated ranging from 0.0001 to 0.1092 mg $O_2 \cdot L^{-1} \cdot d^{-1}$ (Fig. 3). The CR was higher at the surface layer than
the corresponding bottom layer in all stations (Fig. 3, Table S3). The CR at surface layer ranged from 0.22 to 1.68 mg $O_2 \cdot L^{-1}$
$^{1} \cdot d^{-1}$, and that at bottom layer ranged from 0.002 to 0.82 mg $O_2 \cdot L^{-1} \cdot d^{-1}$ (Fig. S4). Based on the ratio between NOD and CR,
nitrification contributed 0.01-17.82 % and 0.009-181.91 % of total oxygen consumption at the surface and bottom layer,
respectively (Fig. 3). It is noteworthy that nitrification contributed substantially to the total oxygen consumption in the upper
estuary and bottom hypoxic water. For the upper estuary in Lingdingyang, nitrification potentially contributed 6.18 % and
9.45 % of the total oxygen consumption at station A01 and A05, respectively. As for the bottom hypoxic water, nitrification
accounted for 28.14 % at F101, 11.28 % at F301, 8.15 % at F303, 4.53 % at A09, 64.89 % at F305 and 181.91 % at F701 of
the total oxygen consumption.

## 3.3 Spatial patterns of the abundance of AOA and β-AOB

As inferred from the *amoA* gene copy number, AOA were 2-3 orders of magnitude more abundant than β-AOB (Fig. 4, Table
S2). The archaeal *amoA* gene was more abundant at the bottom layer than at the surface layer (Fig. 5). The abundance of
archaeal *amoA* gene ranged from $6.27 \times 10^4$ to $3.63 \times 10^7$ copy$\cdot L^{-1}$ at surface layer and $3.59 \times 10^5$ to $4.98 \times 10^8$ copy$\cdot L^{-1}$ at the
bottom layer, with maximum abundance occurred at the bottom layer of station F405. The archaeal *amoA* gene abundance
showed a general decreasing trend from the upper estuary to the continental shelf at the surface layer (Fig. 4 and 5, Table S2).
It is noteworthy that archaeal *amoA* gene was highly abundant in the hypoxic water located in the lower reach of the estuary.
The abundance of β-proteobacteria *amoA* gene at surface layer ranged from $2.03 \times 10^2$ to $1.07 \times 10^5$ copy$\cdot L^{-1}$, while it ranged
from $1.91 \times 10^3$ to $2.44 \times 10^5$ copy·$L^{-1}$ at the bottom layer (Fig. 5, Table S2). The β-proteobacteria *amoA* gene abundance
peaked at the surface layer of station A01 in the upper estuary of Lingdingyang with $1.07 \times 10^5$ copy·$L^{-1}$ while the lowest
abundance was detected at the surface layer of station A12 with $2.03 \times 10^2$ copy·$L^{-1}$. In general, the spatial pattern of β-
proteobacteria *amoA* gene at the surface layer was more abundant at the upper estuary of Lingdingyang (station A01, A05 and
Modaomen (station F303), while the abundance decreased seaward at the bottom layer. Overall, the AOA showed higher
abundance in the free-living fraction while AOB was more abundant in the particle attached fraction (Fig. 5, Table S2). At
RNA level, archaeal *amoA* gene ranged from $6.03 \times 10^2$ to $3.21 \times 10^6$ copy·$L^{-1}$ while β-proteobacteria *amoA* gene were under
detection limit (Table S4). Nitrification rate showed a positive correlation with the total abundance of β-AOB ($r_s$= 0.38, P <
0.05) at DNA level. At the particle attached fraction, nitrification rate displayed positive correlations with the abundance of
AOA ($r_s$= 0.38, P < 0.05) and β-AOB ($r_s$= 0.33, P < 0.05), respectively.

**3.4 Phylogenetic diversity of AOA**

Given that the AOA were the dominant ammonia oxidizers throughout the estuary, we further investigated the phylogenetic
diversity of AOA at DNA and RNA levels in 13 stations covering from the upper estuarine to shelf environments (Fig. 6, 7
and 8). In total, 191,748 high-quality *amoA* sequences were retrieved from 76 samples in the 13 stations (Table S5). OTUs
were detected at 95 % DNA similarity after removal of singletons and doubletons. Top OTUs (OTUs with mean relative
abundance ≥ 0.1 % among all samples) were focused in this study. The Maximum likelihood (ML) phylogenetic tree showed
that the top 85 OTUs affiliated to WCA sublineages and SCM1-like clade according to the reference sequences in Jing et al.
2017 and Cheung et al. 2019 (Jing et al. 2017; Cheung et al. 2019). More than half of the top OTUs were affiliated to the two
WCA sublineages, WCA I (13 OTUs) and WCA II (32 OTUs). Besides, diverse OTUs that affiliated to the SCM1-like clade,
which showed > 90 % DNA similarity with the *amoA* sequences of *Nitrosopumilis maritimus* SCM1, were recovered. These
SCM1-like OTUs were grouped into four sublineages according to the topology of the ML tree, includes SCM1-like-I (10
OTUs), SCM1-like-II (16 OTUs), SCM1-like-III (6 OTUs) and SCM-like-IV (8 OTUs) (Fig. 6 and 7). The SCM1-like-III
were also phylogenetically close to *Nitrosoarchaeum limnia* (Fig. 6, 7 and S2)

**3.5 Differential distribution of AOA sublineages at DNA and RNA level**

As revealed by the NMDS plot, a strong dissimilarity between DNA and RNA communities were observed (Fig. 8). Different
AOA sublineages showed distinct distributional patterns (Fig. 6, 7 and 8). WCA I was mainly distributed in bottom layers
except for the upper reach of Lingdingyang. At the surface layer, WCA I was generally a minor component of the AOA
community, though it was dominant occasionally in the plume area with intermediate salinity. At RNA level, WCA I showed
low relative abundance in the surface layer with mid salinity and an increasing trend seaward (Fig. 6, 7 and 8).
The AOA community at DNA level was dominated by WCA II which showed a ubiquitous distribution across the whole
salinity range of 0.1-34.7. Exceptionally, WCA II was outnumbered by SCM1-like-III at the surface layer at station F301 near

the Modaomen and Huangmaohai close to freshwater discharge. At RNA level, WCA II showed similar distributional patterns and relative abundance with WCA I sharing an increasing proportion of the active AOA community from the upper estuary to the continental shelf (Fig. 6, 7 and 8).

SCM1-like sublineages were surprisingly dominating the active AOA communities at RNA level expect SCM1-like III, which was dominating at stations near river outlets. Among SCM1-like sublineages, the SCM1-like-III was the most abundant at DNA level. Their distribution was limited to surface water of the Pearl River and freshwater plume (salinity < 14) (Fig. 6, 7 and 8). The distribution of SCM1-like-III at RNA level was limited to the freshwater regions (Fig. 6, 7), similar to its distribution pattern showed at DNA level. In addition, SCM1-like-III was the least abundant among the SCM1-like sublineages at RNA level. SCM1-like-I distributed mainly at the lower reach of the estuary. The SCM1-like-II dominated the active AOA communities in the Pearl River and its lower reach at the bottom layer, while the SCM1-like-IV showed high relative abundance at the surface layer (Fig. 8). The SCM1-like-I was less abundant than SCM1-like-II at RNA level at the bottom layer, and its spatial pattern was similar to SCM1-like-II.

**3.6 Correlation between AOA sublineages and environmental factors**

To reveal the connections between the relative abundance of AOA sublineages and environmental factors, correlations between different sublineages and environmental factors were examined using Spearman correlation coefficients. The AOA communities were separated into 4 parts: surface DNA, surface RNA, bottom DNA, and bottom RNA levels, and were analyzed with the corresponding environmental factors. Generally, the relative abundance of AOA sublineages showed a more significant correlation with environmental factors both at DNA and RNA levels at the bottom layer compared to surface layer (Fig. 10). Among 9 environmental factors, salinity was the most significant factor affecting the distribution of AOA sublineage.

The sublineages of WCA showed a strong positive correlation with salinity while SCM1-like sublineages showed a negative correlation with salinity. At RNA level in the bottom layer, SCM1-like-I and IV were positively correlated with nutrient concentration and non-phototropic prokaryotic cell abundance while negatively correlated with salinity and dissolved oxygen concentration. SCM1-like-III showed a strong negative correlation with salinity at both surface and bottom layers. In general, WCA sublineages were negatively correlated with nutrient concentration, while SCM1-like sublineages were positively correlated with nutrient concentration. Ammonium showed no significant correlation with AOA sublineages.

The Spearman correlation between nitrification rates and the relative abundance of AOA sublineages in RNA based communities were also tested (Fig. 10). SCM1-like-III showed a positive correlation ($r_s$= 0.72, P < 0.05) with nitrification rate at surface water, while SCM1-like-I ($r_s$= 0.81, P < 0.05) and SCM1-like-IV ($r_s$= 0.73, P < 0.05) sublineages showed positive correlations with nitrification rates at the bottom layer. Besides, WCA I showed a positive correlation with nitrification rates ($r_s$= 0.75, P < 0.05) only at the surface layer, while WCA II showed a negative correlation ($r_s$= -0.73, P < 0.05) with nitrification rates at the bottom layer.

## 4 Discussion

### 4.1 Nitrification and its oxygen consumption in the hypoxia zone

We observed a widespread hypoxia-zone at the lower estuary of Pearl River, extending from Huangmaohai to south of Hong Kong which was a result of both physical and biogeochemical conditions (Fig. 2f). During the 2017 summer cruise, river discharge was high as indicated by the salinity at the surface layer (Fig. 2a), which is the typical wet season pattern of Pearl River estuary (Harrison et al. 2008). The continuous river discharge sustained strong water column stratification at the lower estuary which prevents the efficient supply of oxygen to the bottom water. Furthermore, a high concentration of nutrients associated with the freshwater from three sub-estuaries sustained high phytoplankton biomass in the lower reach of the estuary (Fig. 2b). The massive locally generated and riverine organic matter sunk down to the bottom layer and they were rapidly degraded by heterotrophic prokaryotes, resulting in high oxygen consumption (Harrison et al. 2008; Lu et al. 2018).

Our results suggest that nitrification could contribute a large proportion of oxygen consumption in the hypoxia zone (Table S3). Despite limited data with large variation, our estimate falls in general the ranges of previous reports. In the eutrophic Delaware River estuary, nitrification accounted for over 20 % of the oxygen consumption river downstream (Lipschultz et al. 1986). Intensive nitrification was observed at intermediate salinities, and it accounted for 20 to over 50 % of oxygen consumption in the Mississippi River plume (Pakulski et al. 1995). In the downstream of Pearl River (from Guangzhou to Humen), nitrification could contribute to one-third of total oxygen consumption (Dai et al. 2008). In our study, high community respiration rates as well as nitrification rates were observed at lower reach of the Pearl River estuary corresponding to the hypoxia zone at the bottom layer (Fig. 2f). It is well-known that ammonia, the substrate of nitrification, was produced during the organic matter degradation (respiration) (Ward, 1996). Thus, high rate of nitrification was supported not only by riverine ammonia but also by rapid organic matter degradation. We observed the high nitrification rate associated with the upper estuary and hypoxia zone (Fig. 3). Respiration and nitrification are both important and coupled oxygen-consuming processes. Comparing with the community respiration, we found that nitrification contributed a substantial proportion (averaged 12.18 %, excluding the unusual number of 181.91 % from F701) to total oxygen consumption at the bottom layer. We found that the NOD exceeded CR at the bottom layer of station F701, which might be caused by the underestimation of CR in oxygen depleted condition using the traditional incubation and titration method. Sampou and Kemp have found that oxygen concentration is one of the limiting factors of CR. In their study, CR was found to decrease when DO was lower than $0.8 \text{ mg} \cdot \text{L}^{-1}$ (Sampou and Kemp 1994). Nitrification can remain active under nanomolar range of oxygen (< 10 nM) (Bristow et al. 2016). During the cruise, the lowest oxygen concentration was $0.54 \text{ mg} \cdot \text{L}^{-1}$ (16.88 μM) which would not limit the nitrification activities (Bristow et al. 2016). Hence, in the Pearl River Estuary, nitrification could substantially draw down oxygen concentration and sustain hypoxia formation at the lower estuary. It should be mentioned that exceedance of potential NOD over the total oxygen consumption was also found in the Changjiang estuary by Hsiao et al. (2014), and they speculated that other oxidants (Fe and Mn) could oxidize ammonia.

## 4.2 Relative distribution of AOA and AOB in Pearl River Estuary

Both AOA and AOB are present in estuarine environment, however, their corresponding contribution to nitrification activities remained under explored. It has been well identified that AOA outnumber AOB by orders of magnitude in pelagic waters, whereas in the estuarine environments, the ratios of AOA and AOB were rather variable. Based on qPCR of *amoA* gene, AOB were more abundant than AOA in many coastal and estuarine sediments (Caffrey et al. 2007; Mosier and Francis 2008; Santoro et al. 2008; Magalhaes et al. 2009; Wankel et al. 2011), while AOA were orders of magnitude more abundant than AOB in other estuaries and coastal environments (Caffrey et al. 2007; Moin et al. 2009; Abell et al. 2010; Bernhard et al. 2010; Mosier and Francis 2011). The variance and relative importance of AOA and AOB, as well as the nitrification rates in estuarine environments were shown being related to physicochemical parameters such as salinity, dissolved oxygen, ammonia and pH (Bernhard and Bollmann 2010; Mosier and Francis 2011). Comparing to the previous estuarine studies based on DNA survey, we conducted comprehensive quantification of AOA and β-AOB abundance at both DNA and RNA levels, in association with in situ nitrification rates measurements in the Pearl River estuary. In Pearl River estuary, AOA outnumbered AOB throughout the estuarine at DNA level. At RNA level, AOA was detectable, but AOB was not, suggesting that AOA were the active ammonia oxidizers in the Pearl River estuary. Moreover, size-fractionated study revealed that AOA were mainly distributed in the free-living fraction, while AOB were associated with the particles near upper estuary (Fig. 5 and Table S2), which may be explained by higher substrate (ammonia) concentration requirement of AOB than AOA (Martens-Habbena et al. 2009).

## 4.3 Unneglectable disagreement of the AOA community at DNA and RNA level

In our study, the positive correlations between nitrification rates and different AOA sublineages suggested the divergence of nitrification activities among the AOA community in the dynamic estuarine ecosystems (Fig. 10). Given that AOA plays a central role in the nitrogen cycle, the physiological characteristics of the highly diverse AOA are an essential basis for understanding the nitrogen cycle in the current and future ocean. With the limitation of underrepresented cultures and genomes, numerous AOA related studies in the ocean were based on amplicon sequencing and qPCR targeting archaeal *amoA* (Beman et al. 2008; Bernhard and Bollmann 2010; Peng et al. 2013; Santoro et al. 2017; Alves et al. 2018). However, it should be noted that almost all these studies were based on DNA samples. In our study, the obvious disagreement between the AOA communities at DNA and RNA levels (Fig. 8) indicated that different AOA sublineages may have functional differences. Coincidentally, a similar phenomenon has also been recently reported in the terrestrial ecosystem, in which *Nitrososphaera* and its sister groups were more active than *Nitrosotalea* in acidic forest soils (Wu et al. 2017). In Baltic Sea, a distinct AOA community were retrieved from RNA level and a few phylotypes related to *Nitrosomarinus* showed widespread expression in the coastal region (Happel et al. 2018). As reported in a previous study in the Pacific Ocean, the *amoA* gene abundance of WCA and WCB have no correlation with nitrification rates throughout the water column indicated the active functional group of AOA might be underrepresented in DNA based studies (Smith et al. 2016). In the light of our finding, the abundant AOA sublineages (WCA) can be much less active ammonia oxidizers than the rare sublineages (SCM1-like) (Fig. 8 and 9), which

suggested that the DNA-based observations were insufficient to unravel the major ammonia oxidizers in the ocean.
Furthermore, given that highly diverse sublineages of WCA and WCB have recently been reported in the oceanic waters
(Cheung et al. 2019; Lu et al. 2019), the nitrification activity of different AOA sublineages should be further verified in future
field studies.

**4.4 AOA sublineages and their potential niche in the estuarine ecosystem**

The ammonia-oxidizing archaea in the estuarine water were less studied compared to those in estuarine sediments, oceanic
waters, and soils since the discovery of AOA (Damashek et al. 2016). In the sediment of San Francisco Bay, Mosier and
Francis (2008) had proposed a cluster of AOA phylotypes potentially adapted to the low salinity environment (Mosier and
Francis 2008). However, these phylotypes were then also observed in a salt marsh (Moin et al. 2009) which leads to
questionable the low-salinity adaption assumption (Bernhard and Bollmann 2010). On the other hand, exploration of diversity
and biogeography of different AOA were limited by low-coverage clone library method as well as the underrepresented active
population at RNA level. Furthermore, in most cases, relatively weak or no correlations were found between nitrification rates
and archaeal *amoA* gene abundances (Bernhard and Bollmann 2010) indicating diverse physiological characteristics among
ammonia oxidizers. The above-mentioned scenarios raise the necessity to study key and active ammonia oxidizers in the
community to understand their contribution in nitrification activities in the field.
In our study, we found niche partitioning among AOA sublineages in the dynamic PRE ecosystem in which the AOA
community is mainly consisted of WCA and SCM1-like sublineages, while WCB is not detected. This pattern is consistent
with the previous studies that show WCA and SCM1-like are mainly distributed in surface water and WCB is limited to deep
mesopelagic waters (Francis et al. 2005; Beman et al. 2008). In a recent study based on the Tara *Oceans* dataset, WCA I
dominated the surface water AOA communities throughout the global oceans (Cheung et al. 2019). In this study, WCA I was
generally minor in the estuary except for the high salinity bottom water intruded from the South China Sea (Fig. 8), which
indicated that WCA I prefer the conditions of oceanic waters. As revealed by the genomic and proteomic information of its
representative culture (*Candidatus Nitrosopelagicus brevis* CN25), the WCA I have a streamlined genome with high coding
density and are ubiquitously distributed in oligotrophic surface ocean (Santoro et al. 2015). In contrast, WCA II was dominant
in the AOA communities throughout our studied region at DNA level (Fig. 8), which agrees with the previous study that its
relative abundance was generally higher in marginal seas (the Gulf of Mexico, the Red Sea, and the Arabian Sea) than in
oceanic waters (Cheung et al. 2019). The present study showed that WCA II outnumbered WCA I in the estuarine ecosystem,
which strongly indicated a niche partitioning between WCA I (oceanic water preferred) and WCA II (coastal water preferred).
Nevertheless, these two WCA sublineages only contributed a small portion of the archaeal *amoA* gene transcripts and did not
show a significant correlation with nitrification rate (Fig. 10), which indicated that they were not the major ammonia oxidizers
in the estuarine ecosystem. Hence, the ecological function of these abundant WCA sublineages in the estuarine ecosystem
should be further explored in future studies.
Regarding the active populations in RNA level, highly diverse SCM1-like OTUs that are highly similar to *amoA* gene of
Nitrosopumilus maritimus SCM1 were recovered in this study (Fig. 6 and 7) (Konneke et al. 2005). In particular, the 4 SCM1-
like sublineages defined in this study displayed distinct distributional patterns: SCM1-like-I and II mainly distributed in the
lower reach of the river; SCM1-like-IV was mainly active at the surface layer in the estuary; SCM1-like-III was limited to
freshwater, implying distinct niche partitioning of the SCM1-like sublineages (Fig. 8). As inferred from the correlation analysis
result, SCM1-like-I was the major active ammonia oxidizer in the PRE water column. The earlier view presumed that AOA
are chemolithoautotrophs that largely rely on ammonia oxidation for energy acquisition. However, increasing evidence
suggested that marine AOA (i.e. *N. maritimus* strains) can utilize organic nitrogen (i.e. urea and cyanate) as the substrates of
nitrification, or utilize organic nutrient (Qin et al. 2014; Kitzinger et al. 2019). Using the stable isotope probing technology,
the utilization of organic matter provided evidences of heterotrophy of AOA in the salt marsh sediment and oceanic
environment (Seyler, et al. 2014; Seyler et al. 2018; Seyler et al. 2019). Hence, it may explain that the high nitrification
activities of the SCM1-like sublineages were facilitated by the enriched and diverse nitrogen sources in estuarine water. Recent
culture-based studies found the physiology of *N. maritimus* was not significantly influenced by salinity changes in the growth
medium (Elling et al. 2015, Qian et al. 2015), which indicated SCM1-like can tolerant to wide salinity range. Furthermore,
SCM1-like-I showed a positive correlation with non-phototropic prokaryotic cell abundance, which, together with high
abundances of AOA and non-phototropic prokaryotic cell in the hypoxic zone, suggest potential interaction and coupling
between organic matter degradation and nitrification activities. On the other hand, SCM1-like-I and II were the major ammonia
oxidizers in the hypoxic waters (Fig. 10), where nitrification contributed significantly to the total oxygen consumption (Fig.
4). Consistently, *N. maritimus* can actively oxidize ammonia and grow under low oxygen conditions (Qin et al. 2017).
The spatial distribution of SCM1-like-III as well as the negative correlation with salinity indicated that SCM1-like-III is
associated with freshwater discharge. The SCM1-like-III was closely related to the *amoA* gene fragment of *Nitrosoarchaeum*
*limnia* which is a low-salinity adapted species (Fig. S2). The functional potential of low-salinity adaptation of *N. limnia* was
further evidenced by genomic information from an enrichment culture (estuarine sediment from San Francisco Bay) (Blainey
et al. 2011). The genome of *N. limnia* SFB1 possessed numerous motility- and chemotaxis-associated genes that might
facilitate their adaptation to the fluctuating estuarine environment (Blainey et al. 2011). Further genomic and metabolic studies
were needed to understand the ecological role of SCM1-like-III in the freshwater discharge.

**5 Data availability**

The *amoA* gene abundance at DNA level from 23 station along with nitrification rates were listed in Table S2. Nitrification
and community respiration and nitrification oxygen demand were listed in Table S3. The *amoA* abundance at RNA (cDNA)
level from 13 stations were listed in Table S4. The complete sequencing dataset was available at NCBI under the Bioproject
number PRJNA610708. Data will be released once the paper is published. The information of the sequencing samples was
listed in Table S5.

## 6 Author Contributions:

HBL conceived the project and revised the manuscript. YHL performed experiments, analyzed the data, interpreted the data and wrote the manuscript. SYC interpreted the data and wrote the manuscript. XMX edited the manuscript. LC and SJK provided nitrification rates data. JPG provided physical profiles of the project. MHD provided nutrient and dissolved oxygen profiles of the project. All the authors provided critical feedback and help shape the research, analysis and manuscript.

## 7 Competing interests:

The authors declare that they have no conflict of interest.

## 8 Acknowledgments

This work was supported by the Research Grants Council (Hong Kong RGC) Theme-Based Research Scheme (T21-602/16-R), the RGC-NSFC Joint Research Scheme (N_HKUST609/15), and the GRF grants (16128416, 16101318). This study was also supported by the Hong Kong Branch of Southern Marine Science & Engineering Guangdong Laboratory (Guangzhou) (SMSEGL20SC01).

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

**Figure 1. Sampling and rates measurement location during the Pearl River estuary cruise in 2017 summer (HMH-Huangmaohai; MDM-Modaomen; HM-Humen; LDY-Lingdingyang). The sampling location information was overlaid on Google Maps (© Google Maps) image using "ggmap" with "ggplot2" in R (D. Kahle and H. Wickham, 2013).**

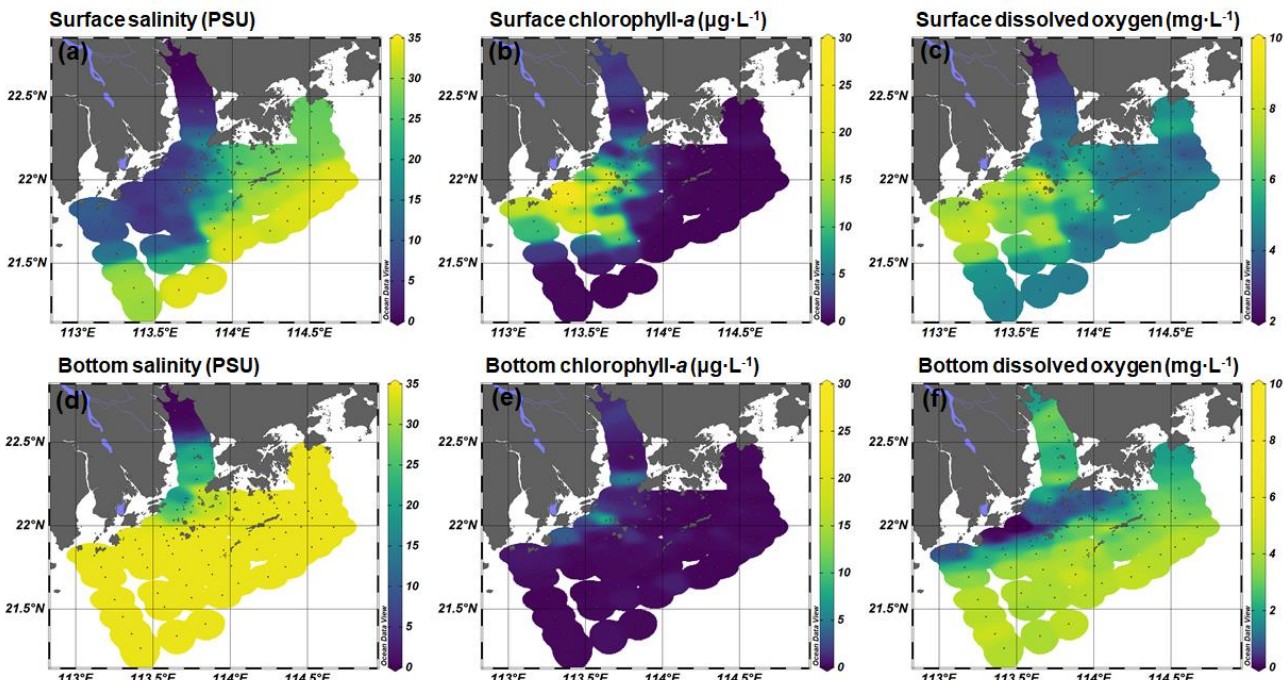


Figure 2. Spatial distribution of (a & d) salinity, (b & e) chlorophyll-*a*, and (c & f) dissolved oxygen concentration at surface and bottom layer during the 2017 summer cruise in Pearl River estuary. These figures were generated using Ocean Data View v. 5.0.0 (http://odv.awi.de).


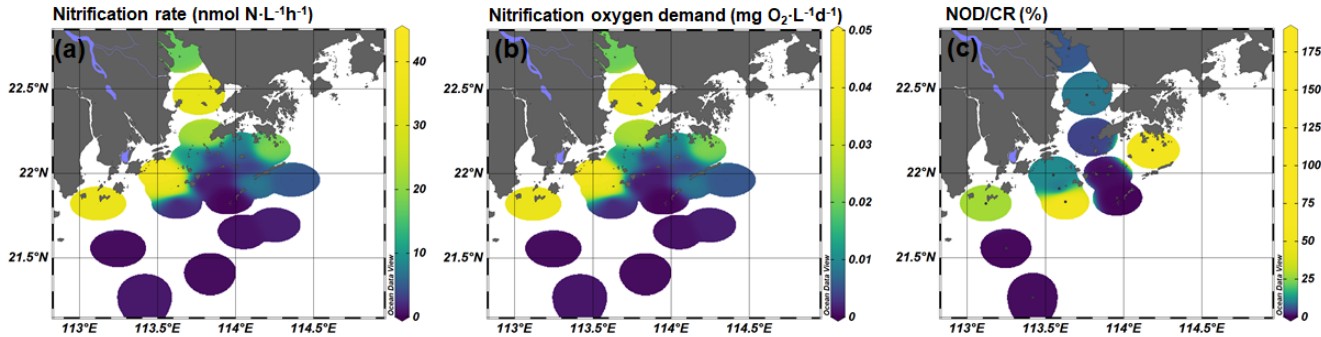


Figure 3. (a) Nitrification rates (nmol N·L$^{-1}$ h$^{-1}$), (b) nitrification oxygen demand (NOD) (mg O2·L$^{-1}$·d$^{-1}$) and (c) nitrification oxygen demand/community respiration (NOD/CR) ratio (%) at the bottom layer.

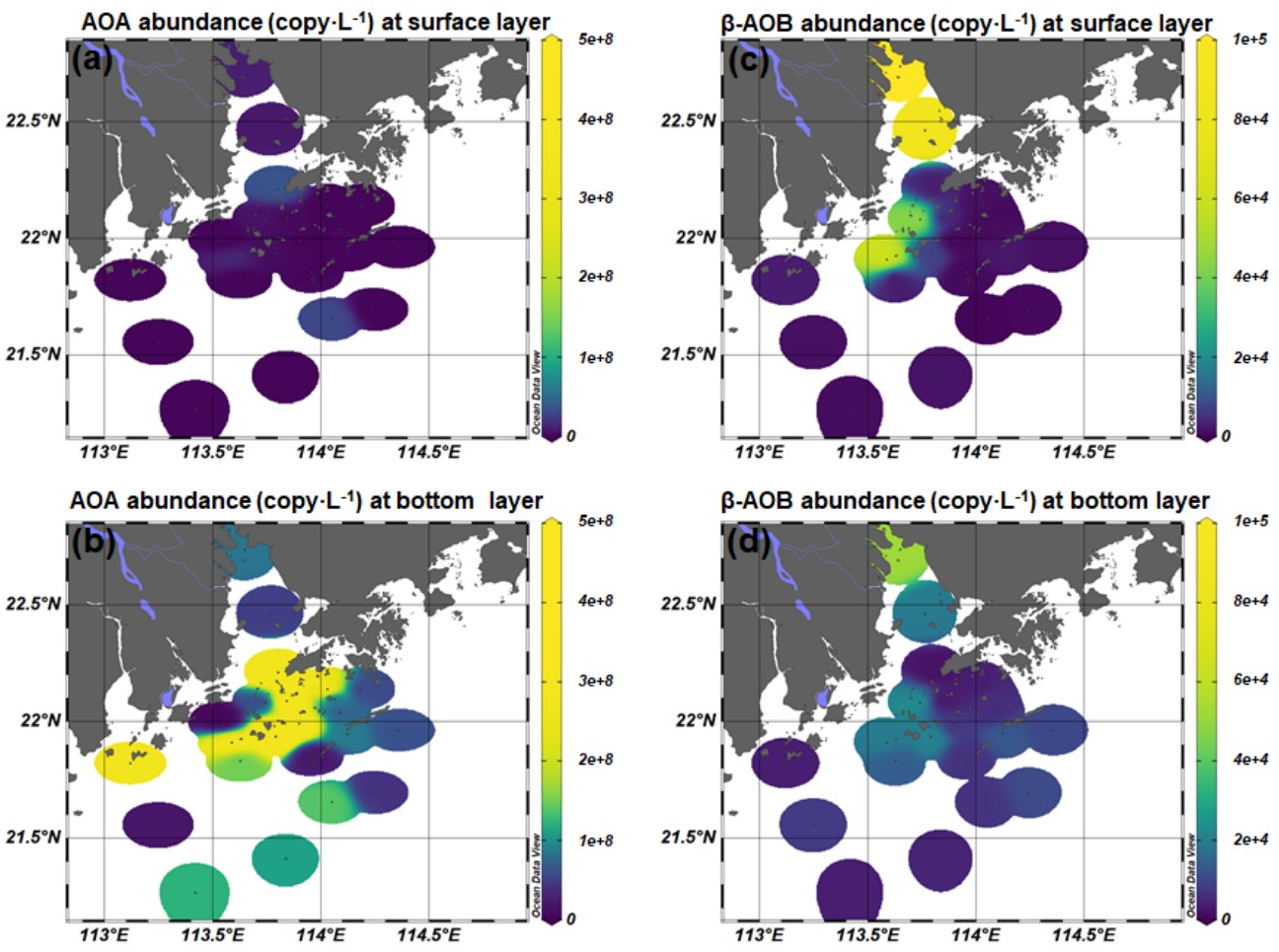

**Figure 4. Spatial distribution of (a & b) AOA and (c & d) β-AOB abundance at the surface and the bottom layer at**
**DNA level.**

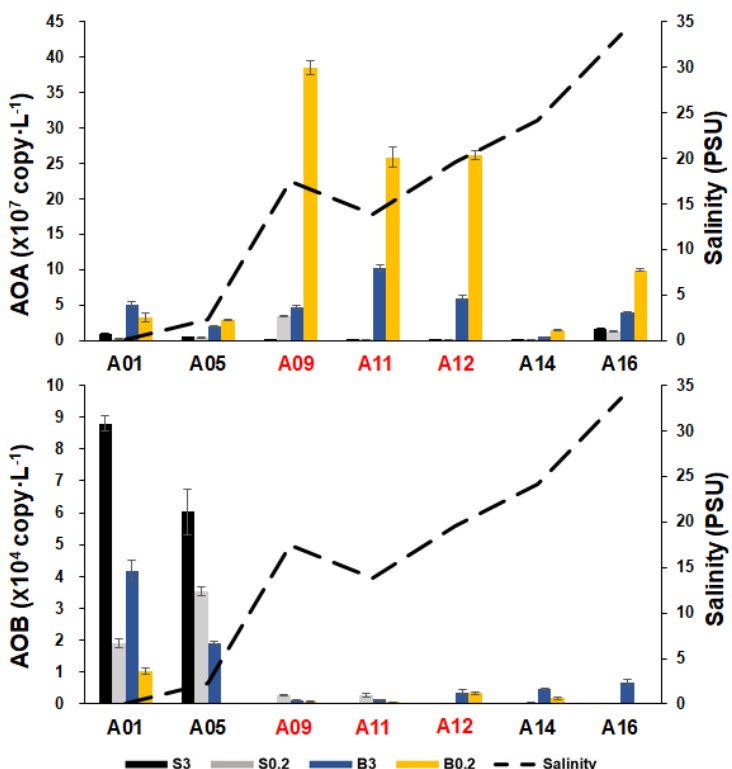

S – surface layer; B – bottom layer;  3 – 3 μm fraction;  0.2 – 0.2-3 μm fraction

**Figure 5. The abundance of AOA and β-AOB at DNA level estimated by qPCR of *amoA* gene along the salinity gradient of the A-transect in the Pearl River estuary. Size fractionation was performed with 3 μm (particle-attached) and 0.2 μm (free-living), and the hypoxic stations (bottom DO < 2 mg·L$^{-1}$) are labelled in red color.**

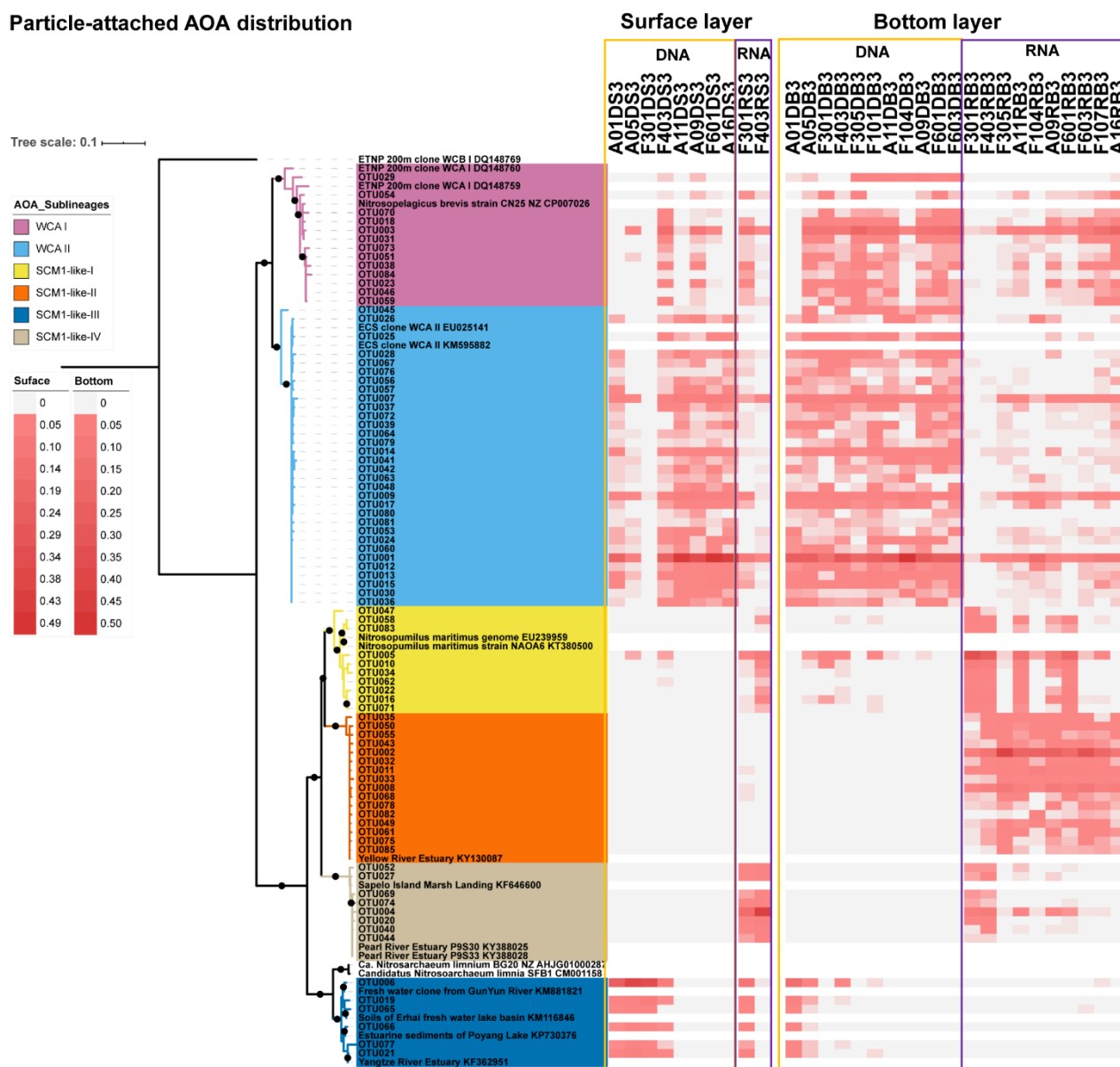

**Figure 6. Maximum likelihood phylogenetic tree of top 85 OTUs based on *amoA* gene sequences using T92+G+I model with 1000 bootstrap. The associated heat map is generated based on the relative abundance of top OTUs in the particle-attached samples. Samples are listed from left to right along the ascending salinity gradient.**

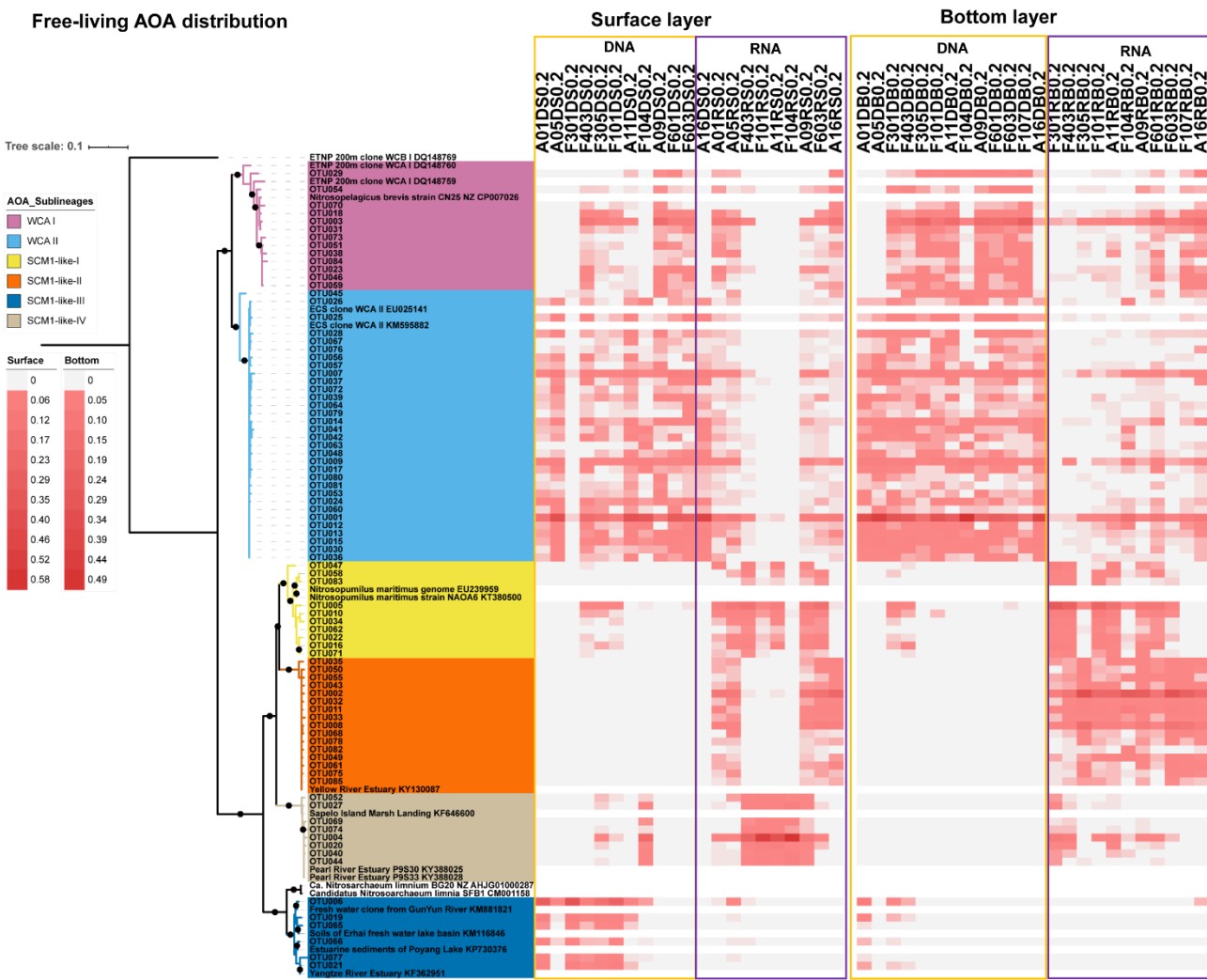


Figure 7. Maximum likelihood phylogenetic tree of top 85 OTUs based on *amoA* gene sequences using T92+G+I model with 1000 bootstrap. The associated heat map is generated based on the relative abundance of top OTUs in the free-living samples. Samples are listed from left to right along the ascending salinity gradient.




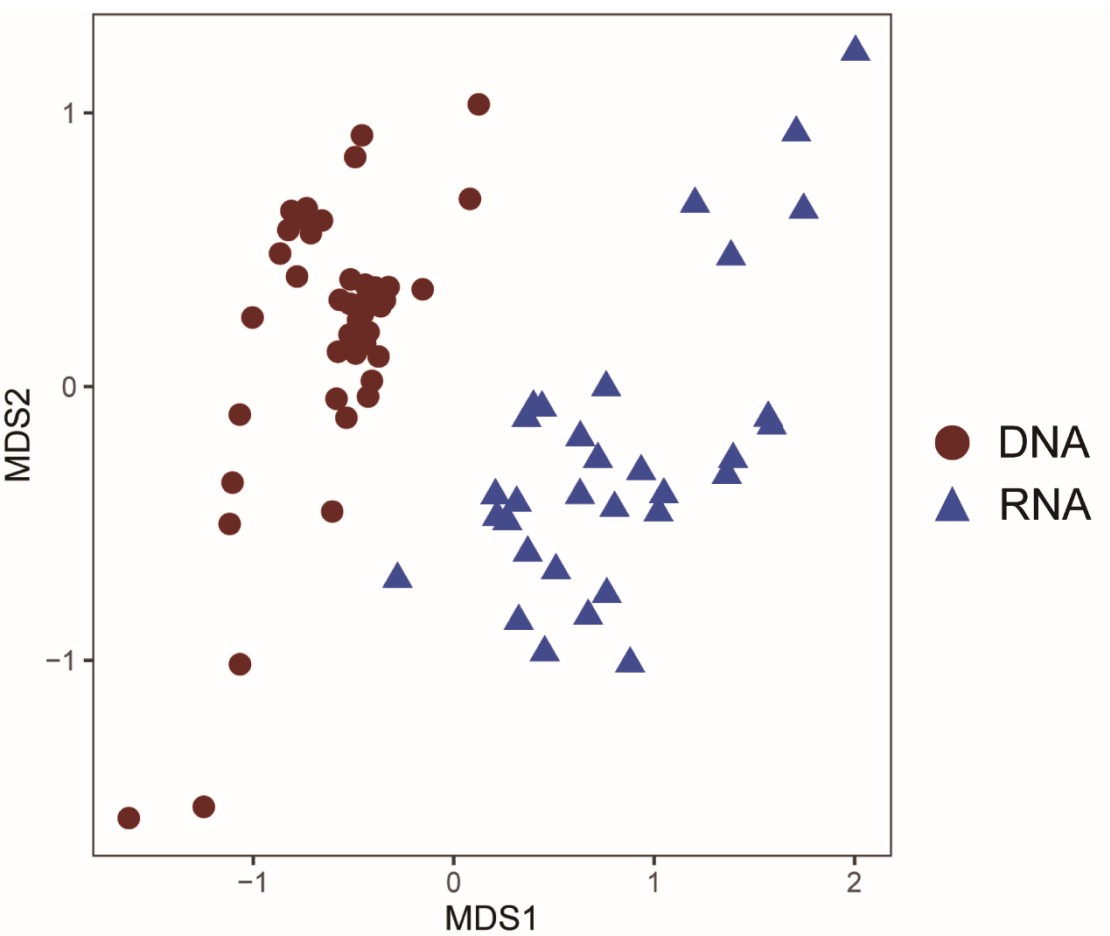


**Figure 8. Nonmetric multidimensional scaling (NMDS) plot of AOA community similarity at DNA and RNA level.**

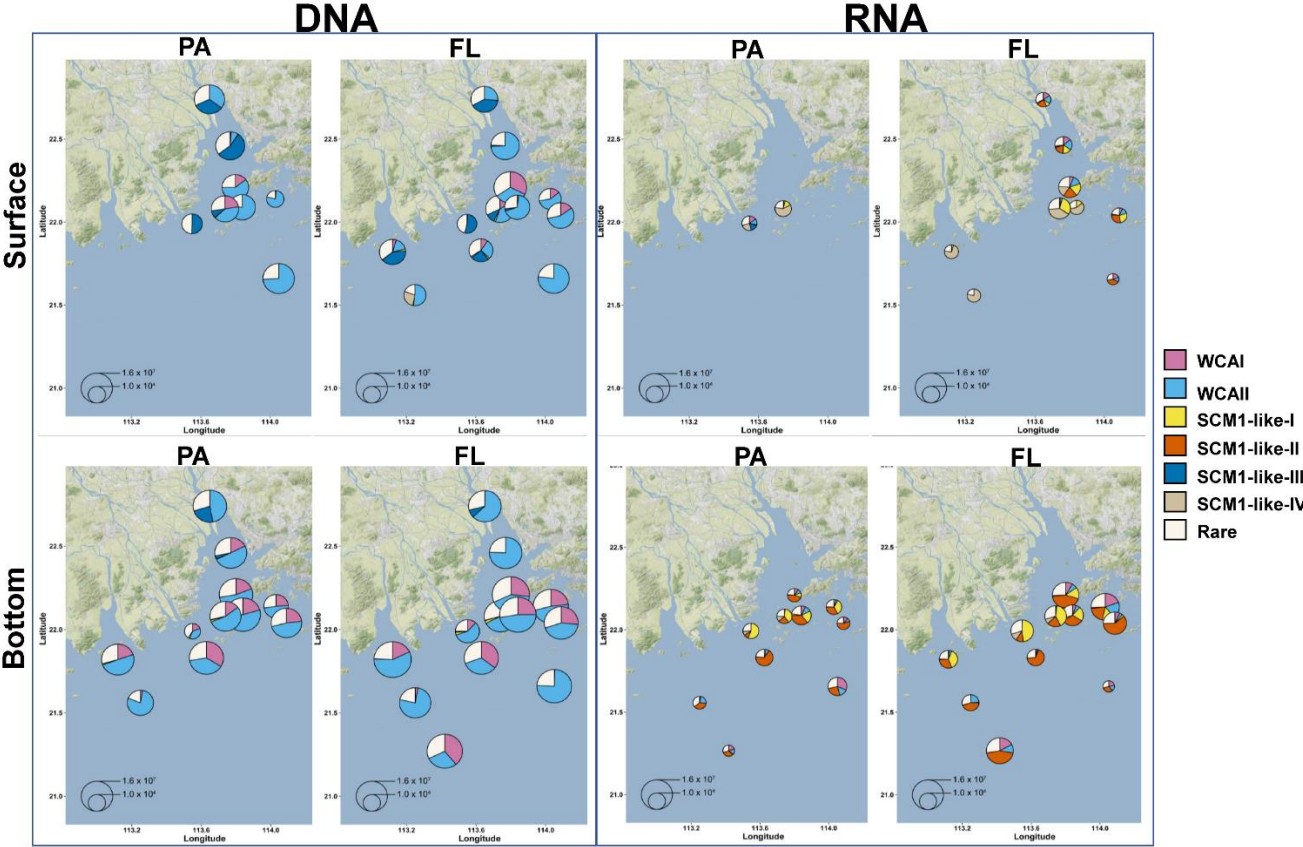


**Figure 9. Free-living (FL) and particle-attached (PA) AOA community composition and distribution in the Pearl River estuary. The size of the pie charts represents the archaeal *amoA* gene abundance quantified by qPCR. For a clear display of the AOA community composition, the minimum size of the pie charts is set as 500 copies·L$^{-1}$. The charts were overlaid on Google Maps (© Google Maps) images using "ggmap" with "ggplot2" in R (D. Kahle and H. Wickham, 2013).**

| Samples | AOA sublineage | Salinity | NR | DO | NH$_4^+$ | NO$_3^-$ | Tem | NO$_2^-$ | Chl-a | NPC |
|---|---|---|---|---|---|---|---|---|---|---|
| Surface_DNA | WCA I | | | | | | | | | |
| | WCA II | | | | | | | | | |
| | SCM1-like-I | | | | | | | | | |
| | SCM1-like-II | | | | | | | | | |
| | SCM1-like-III | | | | | | | | | |
| | SCM1-like-IV | | | | | | | | | |
| Surface_RNA | WCA I | | | | | | | | | |
| | WCA II | | | | | | | | | |
| | SCM1-like-I | | | | | | | | | |
| | SCM1-like-II | | | | | | | | | |
| | SCM1-like-III | | | | | | | | | |
| | SCM1-like-IV | | | | | | | | | |
| Bottom_DNA | WCA I | | | | | | | | | |
| | WCA II | | | | | | | | | |
| | SCM1-like-I | | | | | | | | | |
| | SCM1-like-II | | | | | | | | | |
| | SCM1-like-III | | | | | | | | | |
| | SCM1-like-IV | | | | | | | | | |
| Bottom_RNA | WCA I | | | | | | | | | |
| | WCA II | | | | | | | | | |
| | SCM1-like-I | | | | | | | | | |
| | SCM1-like-II | | | | | | | | | |
| | SCM1-like-III | | | | | | | | | |
| | SCM1-like-IV | | | | | | | | | |

**Figure 10. Spearman correlation between AOA sublineages (relative abundance at DNA and RNA levels) and environmental factors in the surface and bottom layers of the water columns in the Pearl River estuary during summer 2017. Only the significant correlations ($P < 0.05$) are displayed (NR-nitrification rates; DO-dissolved oxygen; Tem-Temperature; NPC-non-phototrophic prokaryotic cells).**