# Peer review of "New Insight to Niche Partitioning and Ecological Function of Ammonia Oxidizing Archaea in Subtropical Estuarine Ecosystem"

_Biogeosciences, 2020_

## Referee Comment (RC1) · Anonymous Referee #1 · 20 Jul 2020

**general comments**

This study by *Lu et al.* provides valuable new insights into the distribution of ammonia-oxidizing archaea (AOA) sublineages and AOA versus ammonia-oxidizing bacteria in the subtropical Pearl River estuary. The study shows a difference in the composition of AOA sublineages at the DNA and RNA level and correlation of nitrification rates with the relative abundance of only one AOA sublineage suggesting a niche partitioning between different AOA sublineages. Furthermore, the authors present data on the contribution of nitrification to oxygen consumption.

Parts of the data set are only superficially mentioned in the manuscript (e.g. fig 8) although they contain valuable information. Especially the comparison between particle attached vs free-living AOA community composition deserves more attention.

NOD/CRs ratios are a central focus of this manuscript. At the same time the NOD rates are part of different manuscript. In order to see the clear separation of focus and content, the other manuscript should be made accessible to the reviewers. This probably would also help to get important information on the method of NOD determination that are missing from this manuscript (e.g. how many time points were taken per rate measurement?).

A lot of emphasis is put on the relative importance of NOD in CR. It is stressed various times throughout the manuscript that NOD is high and at times amounts to more than 200%. However, at these stations NOD is not significantly higher compared to other stations, instead CR rates are VERY low. A critical discussion of the CR rates is absent and should be added to the discussion section. How can the observed patchiness of CR rates be explained?
Furthermore, this raises the question of how well constrained the CR data are. Are they based on two data points per rate measurement? How many replicates have been performed? No standard deviation is reported for NOD or CR. I ask the authors to add this information to the respective tables in the supplementary information and would like them include the number of replicates performed in the material and method section. According to the material and method section, triplicates were performed for the qPCR data. However, standard deviations are also missing in the respective data tables in the supplementary information. I ask the authors to add this.
For the calculation of the inferred nitrification oxygen demand, the authors use improperly balanced equations. This strongly influences the outcome: e.g. for ammonia oxidation, when using

$$NH_3 + 1.5\ O_2 \rightarrow NO_2^- + H_2O + H^+$$

instead of equation (1), the oxygen demand changes by 33%. During carbon fixation, some electrons are used to reduce CO2 and not oxygen. However, the assumption that for every $NH_3$ molecule 1.98 $HCO_3$ gets fixed is hardly realistic. Furthermore, the authors assume 1:1 coupling between ammonia oxidation and nitrite oxidation. However, no data on the abundance of nitrite oxidizers is provided and the rate measurements provided do not distinguish between nitrite or nitrate production. I suggest that the estimate of oxygen demand should focus on the first step of nitrification only or at least a paragraph needs to be added to the discussion section.

The grammar and language need to be revised. There are too many issues throughout the manuscript to list here, which at times makes it hard to follow the authors line of thought.

**specific comments**

l. 63 they would not have overlooked them, but rather underestimated their activity and relative contribution to ammonia oxidation.

ll. 86-87 microbial instead of bacterial.

l. 96 clarify "running seawater"

l. 158 please provide an overview over the 76 samples (which stations and depths are they from) and refer to table S5.  The 2523 reads per file does not match the data reported in table S5. The sample categories provided in table S5 need further explanations.

l. 162 Ion torrent is known for introducing homopolymers. Filtering reads with >8 homopolymers is quite a weak setting considering your aim of "performing fine-scale phylogenetic classification". Please comment.

ll. 170ff. What is the sampling depth of the samples you classified as "bottom".

l. 330 substrate requirement: do the authors mean substrate concentration?

l. 355 "questionable" How so? Such a statement needs to be accompanied with an explanation.

Section 4.1 repeats results in great detail that are already described in the result section. Consider condensing this section.

Fig. 2: figure 2 consists of a selection of graphs to show the most interesting pattern among the environmental parameters measured. This is alright, but the rest of the graphs needs to be provided as well (e.g. supplementary info). For example, surface nitrate concentrations and bottom nitrite concentrations are shown, but bottom nitrate concentrations and bottom salinity are missing.

Fig. 3c: Data are only plotted for a fraction of the stations compared to 3a and b. Why is a part of the data missing?

Fig. 4: please provide the scale in the same number format for AOA and AOB. In order to compare abundances between surface layer and bottom layer please use the same range for the scale for 4a and c and b and d respectively.

Fig. 9: you include the temperature in the Spearman correlation in this table. Therefore, you should also provide the temperature data. Maybe add them to table S2.

Fig.9 and l. 391: How did you quantify heterotrophic bacteria? With the cell quantification method, you reported in the material and method section heterotrophic microbes cannot be distinguished from autotrophic non-phototrophic microbial cells (such as the nitrifiers that this study focuses on).

**technical corrections**

As pointed out above, there are too many issues throughout the manuscript to address here. Some selected comments:

l. 42 "Based on the" instead of "as revealed by"

l. 47 The WCA, WCB, and SCM1-like groups correspond...

l. 102 introduce the abbreviation CR in line 93

Fig. 9: this is a table not a figure. Typos in the first column: Surface.

---

## Referee Comment (RC2) · Anonymous Referee #2 · 30 Jul 2020

I feel that this manuscript contains valuable information regarding ammonia oxidizing archaea in estuarine systems, particularly in that it focuses on processes occurring in the water column rather than the sediment, which, as the authors point out, is under-studied. However, there are numerous issues with the manuscript in its current form.

First and foremost, there are serious issues throughout the manuscript with grammar and syntax. Sometimes these issues are so severe that they obscure the meaning of the text. This made it difficult to grasp the authors' meaning and to review the manuscript effectively.

In general, the description of the methods is unclear and lacking in detail. For example:

[Figure]

line 78: "the 10-50m by 10m interval" What does this mean? lines 87-89: "Sea water was prefiltered... analysis (Liu et al. 2014)." Which analysis was this performed for? line 93: "Community respiration rates were measured" in what? Microcosms? Incubations are mentioned but no volume is given, whether a headspace was left in the bottle... line 94: "running seawater" Outside the (unmentioned) bottle? line 95: "less 10%" Does this mean "less than 10%"? line 96: "The del-15N in NO- x the product of nitrification" I have no idea what this means. line 97: "denitrifier method" What is that? The authors provide citations but for methods but do not explain what they are or how they are performed. Similarly the measurement of the nitrification rate is not described, only cited in an unpublished manuscript. lines 110-111: "Fast DNA SPIN Kit for Soil" Why would you use a soil kit for filter samples from seawater? line 117: "transpired" I assume you mean "transferred" line 136: "the DNA mixture" I don't know what is meant by this. DNA and cDNA?

Because the methods were so unclear in general, it is difficult for me to assess whether the claims made in the results and discussion sections are to be believed. For example, AOA and AOB copy numbers are referred frequently as evidence of dominance of one group over the other. Is this a rational claim, particularly without 16S data to support it? How many copy numbers of the amoA gene do AOA have vs AOB? And if archaeal amoA transcripts are more abundant than bacterial amoA transcripts, does that mean the archaea are more abundant or simply more active? Is the difference is gene/transcript number statistically significant?

As for the measurement of nitrification rates, so little detail is given regarding how these numbers were reached, as to render the data meaningless.

The sections on spatial distribution were in general unclear and difficult to follow.

More specific comments:

line 223: "B-proteobacteria amoA were under detection limit" Not in all your samples though, judging by Figure 5? line 257: "Besides" Besides what? What is meant by

this? line 270: "heterotrophic bacteria abundance" How was this determined? It's not described in the methods. lines 271-272: "Nutrient concentration showed an opposite pattern comparing with salinity" I have no idea what this means. line 274: "which may be introduced by" Again, no idea. lines 295-296: "Intensive nitrification... oxygen consumption (Pakulski et al. 1995)." Was that observed in this study or in the study cited? lines 300-301: "It is well known... organic matter degradation (respiration)." Be that as it may, you still have to cite it- and it's hardly proof that ammonia is supplied to nitrification by this process. line 305: 229.21% oxygen consumption? How do you consume more than 100% of something in a closed microcosm? lines 328-329: "Though size-fractionated... were observed." I don't understand what is meant here. line 330: "higher substrate requirement" of what substrate?

In multiple locations in the document the authors mention previous DNA-based studies of AOA and how such studies may overlook active AOA populations. To begin with, those populations would not be overlooked, but perhaps underrepresented in the data. Additionally, several culture-independent studies of AOA activity utilizing stable isotope probing (in particular, the use of urea as a substrate, and heterotrophy) have been performed in both salt marsh sediment (Seyler et al., 2014, ISME J) and the open ocean (Seyler et al., 2018, FEMS Microbiol Ecol; Seyler et al., 2019, Frontiers Mar Sci), and none of these studies are cited in the text. AOA activity has also been previously described in an estuarine water column using similar techniques to this manuscript (Horak et al., 2013, ISME J; Happel et al., 2018, Env Microbiol)- these should be cited in the text.

As for the figures:

Figure 6 is impossible to read. Could it be separated into two figures by size fraction? Otherwise there's just too much going on.

Figure 7 has me completely puzzled. Firstly because the figure has no axes or scale. Secondly because there's no description of how NDMS analysis was performed in the

[Figure]

text. But most importantly, how is it possible that there is absolutely no overlap between the DNA and RNA sequences? I find this incredibly difficult to believe. Are the DNA and RNA sequence data even capturing the same community?

Figure 8 I think is very interesting, but some of the pie charts are so small as to be illegible.

Figure 9 contains some of the most interesting data in the paper, but the figure needs improvement. I think you could combine this heatmap with your phylogenetic tree, and move Figure 6 to supplemental.

Overall I believe the findings presented in this manuscript are likely of interest to the community. The correlations of various AOA lineages to geochemical data and sampling location are very interesting, if difficult to parse in the manuscript's current format. But the issues with the methods in particular and the text in general made it difficult to understand the findings, and some of the claims lack sufficient evidence. I would very much like to see this manuscript again, after significant revisions.

---

## Author Comment (AC1) · 13 Aug 2020

Dear Reviewer,

We appreciate your constructive suggestions that have led to an improvement of the manuscript. We have fully addressed these comments during the revision. To assist your assessment of our revised manuscripts, we have provided point-to-point response (**blue in color**) to each of the comments by reviewers below. The location of the change in the revised manuscript is highlighted in our response.

Sincerely yours,

Dr. Hongbin LIU (Corresponding author, Email address: liuhb@ust.hk)

**Responses to review 1:**

**general comments**

This study by *Lu et al.* provides valuable new insights into the distribution of ammonia-oxidizing archaea (AOA) sublineages and AOA versus ammonia-oxidizing bacteria in the subtropical Pearl River estuary. The study shows a difference in the composition of AOA sublineages at the DNA and RNA level and correlation of nitrification rates with the relative abundance of only one AOA sublineage suggesting a niche partitioning between different AOA sublineages. Furthermore, the authors present data on the contribution of nitrification to oxygen consumption.

*Response: We thank the reviewer for the accurate summary of our study.*

Parts of the data set are only superficially mentioned in the manuscript (e.g. fig 8) although they contain valuable information. Especially the comparison between particle attached vs free-living AOA community composition deserves more attention.

*Response: While comparing the particle-attached and free-living communities, we did not observe significant difference correspondingly (ANOSIM: r=-0.02177, P=0.797, permutation=999). In contrast, we observed large variation of community along the steep environmental gradient in Pearl River estuary at both DNA and RNA levels (ANOSIM: r=0.7142, P=0.001, permutation=999). Here, we provide two heatmap plots for your reference by splitting Figure 6 (new figure 6 & new figure 7 below): New figure 6: Phylogenetic tree and relative abundance (heatmap)of particle-attached AOA. New figure 7: Phylogenetic tree and relative abundance (heatmap)of free-living AOA. Here, the revised figure 6 and new figure 7 show no significant difference. Therefore, we mainly focused on biogeography of different AOA sublineages and the disagreement between DNA and RNA communities.* Page 28-29 Line 643-651

39

40 *(Revised) Figure 6. Maximum likelihood phylogenetic tree of top 85 OTUs based on amoA*
41 *gene sequences using T92+G+I model with 1000 bootstrap. The associated heat map is*
42 *generated based on the relative abundance of top OTUs in the particle-attached samples.*
43 *Samples are listed from left to right along the ascending salinity gradient.*

[Figure]

44

45 *(Newly added) Figure 7. Maximum likelihood phylogenetic tree of top 85 OTUs based on*
46 *amoA gene sequences using T92+G+I model with 1000 bootstrap. The associated heat map*
47 *is generated based on the relative abundance of top OTUs in the free-living samples. Samples*
48 *are listed from left to right along the ascending salinity gradient.*

49

50 NOD/CRs ratios are a central focus of this manuscript. At the same time the NOD rates are part

51 of different manuscript. In order to see the clear separation of focus and content, the other

52 manuscript should be made accessible to the reviewers. This probably would also help to get

53 important information on the method of NOD determination that are missing from this

54 manuscript (e.g. how many time points were taken per rate measurement?).

55 ***Response:*** *We have elaborated the method of rates measurement (showed below) in the revised*

56 *manuscript. We did not conduct rates measurement with multiple time points. The estimation*

57 *of NOD is based on stoichiometric equation ($NH_3 + 1.5\ O_2 \rightarrow NO_2^- + H_2O + H^+$"). This study*

58 *(using qPCR, Ion-torrent sequencing, rates measurement, environmental data) provided a*

59 *comprehensive view of two group of ammonia oxidizers and more importantly, new insight on*

60 *distinct distribution patterns of AOA sublineages at DNA and RNA level in the estuarine*

*environment in 2017 summer cruise. The other study, using two sets of dark ammonia*

*assimilation rates and nitrification rates from 2015 and 2017 cruises, mainly focus on source*

*and sink of riverine ammonium. We think these two studies contain different and separated*

*contents since they only shared a small part of nitrification rates data in 2017 cruise. Here, we*

*provide the title and abstract of Chen L's work for your reference.*

*"Title: Title: **Dark ammonium transformations in the Pearl River Estuary during summer***

*Abstract*

*Growing human activities in recent decades have collectively resulted in large amounts of*

*nutrients export into coastal oceans. As the most reactive nitrogen species, ammonium ($NH_4^+$)*

*plays the critical role in biogeochemical cycles in estuaries and the coastal ocean. In the highly*

*polluted Pearl River Estuary (PRE), $NH_4^+$ predominates to be the energy source for*

*nitrification, and to be the material source for bacteria and phytoplankton to grow. Both above*

*processes are affected by light, yet in opposite ways. Nevertheless, rare studies paid attention*

*to dual $NH_4^+$ transformation processes specifically during dark conditions. By using nitrogen*

*isotope tracer technique, we quantitatively and simultaneously differentiated two distinctive*

*$NH_4^+$ consumption pathways, i.e., $NH_4^+$ oxidation (AOD) and assimilation (AAD) rates,*

*specially under dark conditions along the PRE during the 2015 and 2017 summer cruises when*

*biological activities were the highest. We found the $NH_4^+$ transformations display a bilayer*

*structure with AAD>AOD in almost all the surface waters and vice versa in all bottom waters,*

*suggesting bacteria and phytoplankton (mainly bacteria) control $NH_4^+$ consumption in surface*

*during the night while nitrifiers are the major $NH_4^+$ consumer in the bottom waters. Through*

*redundancy analysis, we found that both processes are mainly driven by $NH_4^+$ in the PRE*

*during summer."*

*Here is the elaborated method of the rates measurement in the revised manuscript:*

*"Community respiration rates (CR) were measured in triplicate in 60ml BOD bottles without*

*headspace through the dissolved oxygen variance before and after 24 h dark incubation*

*submerged in seawater continuously pumped from sea surface. Nitrification were measured by*

*incubating $^{15}NH_4^+$ amended (less than 10 % of ambient concentration) seawater in duplicated*

*200 ml HDPE bottles in dark for 6-12 h, with temperature controlled by running seawater.*

*After incubation, filtrate (0.2 μm-syringe-filtered) was collected and stored in -20 ℃ for*

*downstream $^{15}NO_x^-$ ($^{15}NO_3^-$ + $^{15}NO_2^-$) analysis (Sigman et al. 2001).*

*The nitrification rates were calculated using the following equation:*

$$AO_b = \frac{(R_t NO_x^- \times [NO_x^-]_t) - (R_{t0} NO_x^- \times [NO_x^-]_{t0})}{t-t0} \times \frac{\left[14_{NH_4^+}\right] + \left[15_{NH_4^+}\right]}{\left[15_{NH_4^+}\right]} \qquad (1)$$

*In equation 1, $AO_b$ is the bulk nitrification rate. $R_{t0} NO_x^-$ and $R_t NO_x^-$ are the ratios (%) of $^{15}N$ in the $NO_x^-$ pool measured at the initial ($t_0$) and termination ($t$) of the incubation. $[NO_x^-]_{t0}$ and $[NO_x^-]_t$ are the concentration of $NO_x^-$ at the initial and termination of the incubation, respectively. $[^{14}NH_4^+]$ is the ambient $NH_4^+$ concentration. $[^{15}NH_4^+]$ is the final ammonium concentration after addition of the stable isotope tracer ($^{15}NH_4^+$). The $NO_x^-$ was completely converted to $N_2O$ by a single strain of denitrifying bacteria (Pseudomonas aureofaciens, ATCC#13985) which lack $N_2O$-reductase activity (Sigman et al. 2001). The converted $N_2O$ was further analyzed using IRMS (Isotope Ration Mass Spectrometer, Thermo Scientific Delta V Plus) to calculate the isotopic composition of $NO_x^-$ (Sigman et al. 2001; Casciotti et al. 2002; Knapp et al. 2005).We analyzed the correlation between nitrification rates and AOA sublineages. Equation 2 was generally considered as the oxidation of ammonia to nitrite. Inferred from the nitrification rates, we estimated the nitrification oxygen demand (NOD) based on equations 2. Inferred from the nitrification rates, we estimated the nitrification oxygen demand (NOD) based on equation 2. We used NOD/CR ratio (percentage) to evaluate potential the contribution of nitrification to total oxygen consumption in the field.*

*$NH_3$ +$1.5O_2 \rightarrow NO_2^-$ +$H_2O$+ $H^+$    (2)"* Page 5 Line 92-111

A lot of emphasis is put on the relative importance of NOD in CR. It is stressed various times throughout the manuscript that NOD is high and at times amounts to more than 200%. However, at these stations NOD is not significantly higher compared to other stations, instead CR rates are VERY low. A critical discussion of the CR rates is absent and should be added to the discussion section. How can the observed patchiness of CR rates be explained?

Furthermore, this raises the question of how well constrained the CR data are. Are they based on two data points per rate measurement? How many replicates have been performed? No

standard deviation is reported for NOD or CR. I ask the authors to add this information to the respective tables in the supplementary information and would like them include the number of replicates performed in the material and method section. According to the material and method section, triplicates were performed for the qPCR data. However, standard deviations are also missing in the respective data tables in the supplementary information. I ask the authors to add this.

*Response: We have added the standard deviation information in Table S2, S3, S4. We also added information in the methodology section that we performed triplicate in community respiration rates measurement. Nitrification rates were measured in duplicates. Both rates were measured only at the end of incubation and we did not perform multi-time-point measurements. We have to admit that the high contribution ratios may be introduced by the underestimation of community respiration rates at low oxygen condition (Sampou and Kemp 1994). Nevertheless, the NOD/CR ratio in our study is to show the potential effect of active nitrification on oxygen consumption in the estuarine system. As the community respiration rates were inhibited but the nitrification rates were not limited at the DO concentrations observed in our survey, it is suggested that nitrification could potentially contribute a large proportion of oxygen consumption under low DO concentration. We have added discussion on community respiration rates in Section 4.1. Page 11 305-308, 315-317*

*Please see the attached and revised version of Table S2, S3 and S4 at the bottom of this file.*

For the calculation of the inferred nitrification oxygen demand, the authors use improperly balanced equations. This strongly influences the outcome: e.g. for ammonia oxidation, when using

$NH_3 + 1.5 O_2 \rightarrow NO_2^- + H_2O + H^+$

instead of equation (1), the oxygen demand changes by 33%. During carbon fixation, some electrons are used to reduce $CO_2$ and not oxygen. However, the assumption that for every $NH_3$ molecule 1.98 $HCO_3$ gets fixed is hardly realistic. Furthermore, the authors assume 1:1 coupling between ammonia oxidation and nitrite oxidation. However, no data on the abundance of nitrite oxidizers is provided and the rate measurements provided do not distinguish between nitrite or nitrate production. I suggest that the estimate of oxygen demand should focus on the

first step of nitrification only or at least a paragraph needs to be added to the discussion section.

The grammar and language need to be revised. There are too many issues throughout the manuscript to list here, which at times makes it hard to follow the authors line of thought.

*Response: We have removed the equation 2 and 3 in the manuscript and changed our NOD calculation based on equation "$NH_3 + 1.5\ O_2 \rightarrow NO_2^- + H_2O + H^+$" (which is now equation 2 in the revised manuscript). The nitrification rates measurement in this study were performed by adding $^{15}N$ labeled ammonium before dark incubation, then collected the filtrate containing $^{15}NO_x^-$. The $^{14/15}Nitrite$ and $^{14/15}Nitrate$ were converted to $N_2O$ by denitrifer method (Sigman et al, 2001). We have elaborated the method of the nitrification rates measurement in the revised manuscript in section 2.2. We now assume each molecule of ammonia consumes 1.5 molecule of oxygen. The NOD and NOD/CR were recalculated based on equation 2 and listed in the revised version of Table S3, description in Section 3.2 and Section 4.1. Page 2 Line 25; Page 8-9 Line 203-215; Page 12 Line 313-317 We have improved the manuscript by reducing the grammar and syntax as well as following the important suggestions from the reviewer. We hope that the current version is much clearer.*

**specific comments**

l. 63 they would not have overlooked them, but rather underestimated their activity and relative contribution to ammonia oxidation.

*Response: We have changed "overlooked" into "underestimated the importance of some active groups in the natural environment" Page 4 Line 62*

ll. 86-87 microbial instead of bacterial.

*Response: We have changed "bacterial" into "microbial" Page 4 Line 86*

l. 96 clarify "running seawater"

*Response: We have changed it into "Community respiration rates (CR) were measured in triplicate in 60ml BOD bottles without headspace through the dissolved oxygen variance before and after 24 h dark incubation submerged in seawater continuously pumped from sea surface". Page 5 Line 93-94*

178

179 l. 158 please provide an overview over the 76 samples (which stations and depths are they from)

180 and refer to table S5. The 2523 reads per file does not match the data reported in table S5. The

181 sample categories provided in table S5 need further explanations.

182 ***Response:*** *We subsampled the sequencing reads based on the number of the sample that*

183 *contains minimum number of reads before OTU clustering. We added abbreviations for sample*

184 *categories under the* Table S5. *The sampling depth information have been added to* Table S2.

185 *Here is revised Table S5:*

186 *(Revised) Table S5. Basic sample information of sequencing samples and corresponding Shannon index,*
187 *Margalef richness.*

| Station | Lon (E º) | Lat (W º) | Sample Cat. | Sequence No. | Shannon index | Margalef richness |
|---|---|---|---|---|---|---|
| | | | A01RS0.2 | 4469 | 4.26 | 42.06 |
| | | | A01DB0.2 | 25484 | 3.70 | 39.66 |
| A01 | 113.65 | 22.74 | A01DB3 | 33527 | 3.73 | 37.25 |
| | | | A01DS0.2 | 28147 | 3.64 | 37.09 |
| | | | A01DS3 | 30179 | 3.68 | 39.3 |
| | | | A05RS0.2 | 10504 | 4.21 | 43.33 |
| | | | A05DB0.2 | 32747 | 3.25 | 33.3 |
| A05 | 113.77 | 22.46 | A05DB3 | 28121 | 4.00 | 40.49 |
| | | | A05DS0.2 | 27297 | 3.33 | 35.85 |
| | | | A05DS3 | 20389 | 3.42 | 33.75 |
| | | | A09RB0.2 | 21803 | 3.78 | 39.07 |
| | | | A09RB3 | 16585 | 3.87 | 41.38 |
| | | | A09RS0.2 | 12693 | 4.14 | 43.61 |
| A09 | 113.80 | 22.21 | A09DB0.2 | 21927 | 4.04 | 37.99 |
| | | | A09DB3 | 21343 | 3.71 | 33.55 |
| | | | A09DS0.2 | 10794 | 4.07 | 29.95 |
| | | | A09DS3 | 25603 | 3.53 | 37.12 |
| | | | A11RB0.2 | 29345 | 4.12 | 43.19 |
| | | | A11RB3 | 26206 | 3.78 | 39.4 |
| | | | A11RS0.2 | 4080 | 3.26 | 28.6 |
| A11 | 113.84 | 22.09 | A11DB0.2 | 24215 | 3.82 | 37.84 |
| | | | A11DB3 | 22422 | 3.72 | 36.47 |
| | | | A11DS0.2 | 20568 | 3.62 | 38.78 |
| | | | A11DS3 | 29216 | 3.18 | 34.89 |
| | | | A16RB0.2 | 20644 | 4.12 | 40.51 |
| | | | A16RB3 | 24676 | 4.01 | 41.43 |
| A16 | 114.05 | 21.66 | A16RS0.2 | 16931 | 3.88 | 39.06 |
| | | | A16DB0.2 | 30526 | 3.31 | 35.74 |
| | | | A16DS0.2 | 31112 | 3.02 | 31.63 |

| | | | | | | |
|---|---|---|---|---|---|---|
| | | | A16DS3 | 28739 | 3.25 | 35.5 |
| | | | F101RB0.2 | 20949 | 3.67 | 38.37 |
| | | | F101RS0.2 | 2523 | 2.61 | 23.22 |
| F101 | 113.12 | 21.82 | F101DB0.2 | 20840 | 3.61 | 30.87 |
| | | | F101DB3 | 15602 | 3.96 | 36.95 |
| | | | F101DS0.2 | 8348 | 3.90 | 35.38 |
| | | | F104RB0.2 | 33200 | 3.60 | 32.74 |
| | | | F104RB3 | 16037 | 3.69 | 31.77 |
| | | | F104RS0.2 | 33670 | 2.22 | 17.82 |
| F104 | 113.25 | 21.56 | F104DB0.2 | 30782 | 2.84 | 28.32 |
| | | | F104DB3 | 30769 | 2.69 | 26.59 |
| | | | F104DS0.2 | 6990 | 3.01 | 30.22 |
| | | | F107RB0.2 | 21167 | 3.89 | 40.88 |
| F107 | 113.42 | 21.27 | F107RB3 | 5633 | 3.89 | 38.1 |
| | | | F107DB0.2 | 20909 | 3.90 | 35.52 |
| | | | F301RB0.2 | 17778 | 3.76 | 34.19 |
| | | | F301RB3 | 16657 | 3.48 | 34.53 |
| | | | F301RS3 | 5653 | 4.03 | 37.6 |
| F301 | 113.55 | 21.99 | F301DB0.2 | 22088 | 3.82 | 38.42 |
| | | | F301DB3 | 3436 | 4.19 | 31.49 |
| | | | F301DS0.2 | 7823 | 3.40 | 27.44 |
| | | | F301DS3 | 20310 | 3.51 | 26.54 |
| | | | F305RB0.2 | 27580 | 3.35 | 36.05 |
| | | | F305RB3 | 27095 | 3.20 | 33.45 |
| F305 | 113.63 | 21.83 | F305DB0.2 | 18856 | 3.96 | 33.86 |
| | | | F305DB3 | 21410 | 3.78 | 35.12 |
| | | | F305DS0.2 | 7007 | 4.20 | 42.21 |
| | | | F403RB0.2 | 10000 | 3.86 | 37.69 |
| | | | F403RB3 | 8858 | 3.69 | 38.31 |
| | | | F403RS0.2 | 4431 | 3.57 | 31.38 |
| F403 | 113.74 | 22.08 | F403RS3 | 4166 | 3.04 | 28.24 |
| | | | F403DB0.2 | 21959 | 3.91 | 40.19 |
| | | | F403DB3 | 21744 | 3.85 | 38.99 |
| | | | F403DS0.2 | 19571 | 4.26 | 43.7 |
| | | | F403DS3 | 20370 | 3.83 | 36.83 |
| | | | F601RB0.2 | 27041 | 4.12 | 43.22 |
| | | | F601RB3 | 22320 | 3.75 | 38.81 |
| F601 | 114.03 | 22.14 | F601DB0.2 | 18421 | 3.82 | 34.78 |
| | | | F601DB3 | 20092 | 3.80 | 33.59 |
| | | | F601DS0.2 | 23411 | 3.70 | 37.44 |
| | | | F601DS3 | 15932 | 2.94 | 33.22 |
| | | | F603RB0.2 | 30619 | 3.55 | 37.54 |
| F603 | 114.09 | 22.04 | F603RB3 | 9410 | 3.55 | 38.81 |
| | | | F603RS0.2 | 5859 | 3.90 | 39.93 |

| | | | |
|---|---|---|---|
| F603DB0.2 | 16912 | 3.96 | 40.71 |
| F603DB3 | 19693 | 3.81 | 35.48 |
| F603DS0.2 | 18314 | 3.78 | 36.1 |

* Sample categories: Station ID + D/R (DNA/RNA) + S/B (Surface/Bottom) + 3/0.2 (Particle attached (>3 μm)/Free-living (3-0.2 μm)).

l. 162 Ion torrent is known for introducing homopolymers. Filtering reads with >8 homopolymers is quite a weak setting considering your aim of "performing fine-scale phylogenetic classification". Please comment.

*Response: The quality control standards resulted that the mean length of homopolymers is 3. The length of the maxhomopolymer in the top OTU sequences we used for phylogenetic analysis in our study is 4, so we think the quality control had excluded error from homopolymers introduced by the Ion torrent.*

ll. 170ff. What is the sampling depth of the samples you classified as "bottom".

*Response: The sampling depth information was added to the revised Table S2.*

l. 330 substrate requirement: do the authors mean substrate concentration?

*Response: Yes, we mean substrate concentration. We have added "concentration".* Page12 Line 339

l. 355 "questionable" How so? Such a statement needs to be accompanied with an explanation.

*Response: In line 361 to 363, the low-salinity adapted cluster were proposed by Mosier and Francis in 2008, however, a later study by Molin in 2009 observed these phylotypes in salt marsh with high salinity, which led to the low-salinity adaptation cluster questionable. This was summarized by Bernhard and Bollmann 2010. We think we had the explanation.*

Section 4.1 repeats results in great detail that are already described in the result section. Consider condensing this section.

*Response: We have removed the repeated results.* Page 11 Line 299-300

215

Fig. 2: figure 2 consists of a selection of graphs to show the most interesting pattern among the environmental parameters measured. This is alright, but the rest of the graphs needs to be provided as well (e.g. supplementary info). For example, surface nitrate concentrations and bottom nitrite concentrations are shown, but bottom nitrate concentrations and bottom salinity are missing.

*Response: We have moved all nutrient plots to the supplementary materials. The current version of figure 2 showed below contains the spatial pattern of salinity, chlorophyll-a and DO concentration at both surface and bottom layer. The nutrient plots of nitrate, nitrite and ammonia were moved to supplementary in Figure S3.* Page 24 Line 627-631; Supplementary Figure S3

226

*(Revised) Figure 2. Spatial distribution of (a & d) salinity, (b & e) chlorophyll-a, and (c & f) dissolved oxygen concentration at both surface and bottom layer during the 2017 summer cruise in Pearl River estuary. These figures were generated using Ocean Data View v. 5.0.0 (http://odv.awi.de).*

[Figure]

231

*(Newly added) Figure S3. Spatial distribution of (a & d) nitrate, (b & e) ammonium, and (c & f) nitrite concentration at both surface and bottom layer during the 2017 summer cruise in Pearl River estuary. These figures were generated using Ocean Data View v. 5.0.0 (http://odv.awi.de).*

Fig. 3c: Data are only plotted for a fraction of the stations compared to 3a and b. Why is a part of the data missing?

*Response: The comparisons were only performed for stations where community respiration rates were measured. We did not conduct the measurements of community respiration rates at many stations as we did for the nitrification rates. The spatial distribution of community respiration rates at the bottom layer was newly added as Figure S4 in supplementary. The citations of these figures were revised accordingly.*

[Figure]

Community respiration rates at bottom layer (mg $O_2 \cdot L^{-1} d^{-1}$)

244

245 *(Newly added) Figure S4. Spatial distribution of community respiration rates at the bottom*

246 *layer (mg $O_2 \cdot L^{-1} d^{-1}$).*

247

248 Fig. 4: please provide the scale in the same number format for AOA and AOB. In order to

249 compare abundances between surface layer and bottom layer please use the same range for the

250 scale for 4a and c and b and d respectively.

251 *Response: We have changed the number format and used same scale range for corresponding*

252 *figures in Figure 4. (new version is attached below and Figure 4 in the main text had been*

253 *replaced with this new version).* Page 26 Line 636-638

[Figure]

254

*(Revised) Figure 4. Spatial distribution of AOA and β-AOB abundance at the surface and the bottom layer at DNA level.*

Fig. 9: you include the temperature in the Spearman correlation in this table. Therefore, you should also provide the temperature data. Maybe add them to table S2.

*Response: We have added "Temperature" in table S2.* Supplementary information Table S2

Fig.9 and l. 391: How did you quantify heterotrophic bacteria? With the cell quantification method, you reported in the material and method section heterotrophic microbes cannot be distinguished from autotrophic non-phototrophic microbial cells (such as the nitrifiers that this study focuses on).

*Response: We admit that flow cytometry method cannot distinguish the autotrophic non-phototrophic microbial cells. We have changed the term in to "non-phototrophic prokaryotic cells" with abbreviation "NPC" in the figure legend in Figure 9.* Page 30 Line663; Page11 Line279; Page 14 Line 401-402

**technical corrections**

As pointed out above, there are too many issues throughout the manuscript to address here. Some selected comments:

l. 42 "Based on the" instead of "as revealed by"

*Response: We have revised "as revealed by" to "Based on the"* Page 3 Line 41

l. 47 The WCA, WCB, and SCM1-like groups correspond...

*Response: We have revised accordingly.* Page 3 Line 46

l. 102 introduce the abbreviation CR in line 93

*Response: We have added abbreviation "CR" in line 93.* Page 5 Line 93

Fig. 9: this is a table not a figure. Typos in the first column: Surface.

*Response: Sorry for the typo. We have corrected it. We considered this heatmap as a figure.*

 *It is now figure 10.*

*(Revised) Figure 10. Spearman correlation between AOA sublineages (relative abundance at DNA and RNA levels) and environmental factors in the surface and bottom layers of the water column in the Pearl River estuary during summer 2017. Only the significant correlations (P<0.05) are displayed (NR-nitrification rates; DO-dissolved oxygen; Tem-Temperature; NPC-non-phototrophic prokaryotic cells).*

*Reference*

*Casciotti, K. L., D. M. Sigman, M. G. Hastings, J. K. Bohlke, and Hilkert, A. : Measurement of the oxygen isotopic composition of nitrate in seawater and freshwater using the denitrifier method., Anal. Chem., 74, 4905–4912, https://doi.org/10.1021/ac020113w, 2002.*

*Knapp, A. N., D. M. Sigman, and Lipschultz, F. : N isotopic composition of dissolved organic nitrogen and nitrate at the Bermuda Atlantic time-series study site, Global Biogeochem. Cycle, 19, https://doi.org/10.1029/2004gb002320, 2005.*

*Sigman, D. M., K. L. Casciotti, M. Andreani, C. Barford, M. Galanter, and Bohlke, J. K. : A bacterial method for the nitrogen isotopic analysis of nitrate in seawater and freshwater, Anal. Chem., 73, 4145–4153, https://doi.org/10.1021/ac010088e, 2001.*

*(Revised) Table S2. Quantitative PCR results at DNA level of both AOA and β-AOB in 23 stations*

| Station | Lon (E °) | Lat (W °) | Layer | Salinity (PSU) | DO (mg·L⁻¹) | Temperature (°C) | Ammonium (nmol·L⁻¹) | Nitrification rate (nmol·L⁻¹·h⁻¹) | AOA-PA (Copy·L⁻¹) | AOA-FL (Copy·L⁻¹) | AOB-PA (Copy·L⁻¹) | AOB-FL (Copy·L⁻¹) |
|---|---|---|---|---|---|---|---|---|---|---|---|---|
| F107 | 113.42 | 21.27 | S-1m | 32.30 | 4.53 | 29.07 | 155.70 | 0.21 | 1.54E+04 ± 1.35E+03 | 7.93E+04 ± 4.04E+03 | 1.81E+02 ±3.02E+01 | 8.05E+02 ±1.04E+02 |
| | | | B-41m | 34.51 | 4.09 | 22.77 | 48.64 | 0.96 | 3.31E+04 ±7.10E+03 | 1.22E+08 ±3.06E+06 | 7.77E+02 ±1.57E+02 | 3.03E+03 ±2.97E+02 |
| F104 | 113.25 | 21.56 | S-1m | 16.69 | 6.80 | 31.01 | ND | 0.14 | 2.92E+04 ±8.54E+02 | 1.27E+05 ±1.27E+04 | 4.90E+02 ±1.11E+02 | 7.56E+02 ±1.60E+02 |
| | | | B-28m | 34.45 | 4.26 | 24.06 | ND | 0.33 | 1.09E+06 ±6.11E+04 | 1.76E+07 ±3.61E+05 | 5.17E+03 ±7.73E+02 | 2.83E+03 ±6.77E+02 |
| F101 | 113.12 | 21.82 | S-1m | 10.20 | 6.38 | 29.29 | 67.03 | 1.18 | 4.20E+04 ±5.67E+03 | 1.19E+06 ±3.79E+04 | 1.11E+02 ±4.40E+01 | 2.57E+03 ±1.87E+02 |
| | | | B-9m | 33.73 | 0.54 | 24.18 | 34.78 | 36.62 | 2.61E+07 ±2.00E+05 | 3.95E+08 ±4.51E+06 | 1.67E+03 ±3.30E+02 | 2.00E+03 ±3.71E+02 |
| F309 | 113.84 | 21.41 | S-1m | 33.91 | 4.47 | 29.74 | 32.41 | ND | 1.24E+03 ±6.11E+01 | 2.67E+05 ±1.08E+04 | 1.31E+02 ±4.05E+01 | 1.35E+03 ±4.02E+02 |
| | | | B-43m | 34.51 | 4.21 | 22.36 | 56.68 | 0.40 | 1.31E+05 ±2.48E+04 | 1.10E+08 ±4.61E+06 | 2.57E+03 ±7.72E+02 | 2.02E+03 ±4.51E+02 |
| F305 | 113.63 | 21.83 | S-1m | 9.04 | 7.08 | 30.52 | 233.66 | 1.84 | 4.83E+04 ±9.26E+02 | 3.21E+05 ±2.04E+04 | 4.77E+02 ±5.88E+01 | 8.42E+02 ±1.01E+02 |
| | | | B-26m | 34.43 | 3.47 | 23.80 | 44.11 | 1.28 | 7.27E+07 ±2.47E+06 | 7.42E+07 ±4.36E+06 | 1.08E+04 ±9.10E+02 | 2.80E+03 ±2.97E+02 |
| F303 | 113.59 | 21.91 | S-1m | 7.54 | 6.82 | 30.14 | 104.01 | 0.48 | 7.55E+06 ±2.29E+05 | 6.09E+06 ±1.17E+05 | 2.89E+04 ±1.95E+03 | 3.42E+04 ±3.47E+02 |
| | | | B-18m | 34.45 | 1.44 | 23.40 | 42.73 | 36.37 | 1.40E+08 ±1.25E+07 | 1.62E+08 ±3.61E+06 | 1.65E+04 ±3.31E+03 | 3.16E+03 ±5.28E+02 |
| F301 | 113.55 | 21.99 | S-1m | 6.70 | 7.67 | 29.12 | 865.79 | 5.20 | 5.80E+04 ±2.19E+03 | 3.29E+04 ±3.53E+03 | ND | ND |
| | | | B-6m | 23.17 | 2.10 | 27.25 | 1423.19 | 41.94 | 5.04E+03 ±1.72E+03 | 3.54E+05 ±3.49E+04 | ND | ND |
| F405 | 113.79 | 21.94 | S-1m | 12.29 | 6.53 | 29.05 | 250.81 | 1.48 | 2.48E+05 ±8.02E+03 | 2.65E+06 ±3.61E+04 | 9.73E+02 ±3.05E+02 | 6.54E+03 ±1.14E+03 |
| | | | B-22m | 34.43 | 2.61 | 23.65 | 34.19 | 1.04 | 5.88E+07 ±2.47E+06 | 4.39E+08 ±1.24E+07 | 1.10E+04 ±2.10E+03 | 1.08E+04 ±1.94E+03 |
| F403 | 113.74 | 22.08 | S-1m | 7.56 | 4.11 | 28.85 | 24.08 | 3.07 | 2.02E+06 ±4.77E+04 | 3.63E+06 ±1.86E+05 | 9.57E+03 ±1.94E+03 | 3.62E+04 ±6.24E+02 |
| | | | B-8m | 22.46 | 1.31 | 26.19 | 24.16 | 9.91 | 1.42E+07 ±7.22E+05 | 3.11E+07 ±1.73E+05 | 7.75E+03 ±7.65E+02 | 1.59E+04 ±1.23E+03 |
| A16 | 114.05 | 21.66 | S-1m | 33.67 | 4.73 | 29.77 | 35.32 | ND | 1.70E+07 ±6.61E+04 | 1.33E+07 ±6.36E+05 | ND | ND |
| | | | B-45m | 34.52 | 4.21 | 22.01 | 111.37 | 0.65 | 3.90E+07 ±2.03E+06 | 9.95E+07 ±1.32E+06 | 6.91E+03 ±9.79E+02 | 2.12E+01 ±7.46E+00 |

| Station | Lon (E °) | Lat (W °) | Layer | Salinity (PSU) | DO (mg·L⁻¹) | Temperature (°C) | Ammonium (nmol·L⁻¹) | Nitrification rate (nmol·L⁻¹·h⁻¹) | AOA-PA (Copy·L⁻¹) | AOA-FL (Copy·L⁻¹) | AOB-PA (Copy·L⁻¹) | AOB-FL (Copy·L⁻¹) |
|---|---|---|---|---|---|---|---|---|---|---|---|---|
| A14 | 113.96 | 21.85 | S-1m | 24.15 | 5.26 | 29.98 | 69.85 | 0.44 | 1.20E+05 ±5.63E+03 | 1.16E+06 ±4.58E+04 | ND | 4.77E+02 ±8.29E+01 |
| | | | B-25m | 34.39 | 4.00 | 24.21 | 355.19 | 0.06 | 5.12E+06 ±1.12E+05 | 1.50E+07 ±1.73E+05 | 4.68E+03 ±4.56E+02 | 1.85E+03 ±2.95E+02 |
| A12 | 113.90 | 21.99 | S-1m | 19.56 | 6.68 | 29.82 | 278.65 | 0.80 | 9.21E+05 ±3.39E+04 | 2.73E+05 ±2.98E+04 | 1.80E+02 ±5.64E+01 | 2.25E+01 ±9.03E+00 |
| | | | B-22m | 34.41 | 2.62 | 26.63 | 56.18 | 1.13 | 6.00E+07 ±3.05E+06 | 2.61E+08 ±6.08E+06 | 3.69E+03 ±7.40E+02 | 3.37E+03 ±5.25E+02 |
| A11 | 113.84 | 22.09 | S-1m | 13.88 | 6.37 | 28.72 | 47.10 | 1.13 | 1.24E+06 ±2.30E+04 | 6.56E+05 ±4.11E+04 | 2.69E+01 ±4.30E+00 | 2.83E+03 ±2.58E+01 |
| | | | B-13m | 32.15 | 0.97 | 24.56 | 120.77 | 2.64 | 1.02E+08 ±4.86E+06 | 2.58E+08 ±1.42E+07 | 1.49E+03 ±6.58E+01 | 6.81E+02 ±3.59E+01 |
| A09 | 113.80 | 22.21 | S-1m | 17.52 | 5.39 | 27.93 | 161.39 | 2.58 | 1.36E+06 ±7.81E+04 | 3.50E+07 ±8.62E+05 | 2.56E+02 ±2.95E+01 | 2.60E+03 ±1.97E+01 |
| | | | B-21m | 33.36 | 1.15 | 24.18 | 91.45 | 22.43 | 4.73E+07 ±2.54E+06 | 3.85E+08 ±9.50E+06 | 1.10E+03 ±2.55E+02 | 8.10E+02 ±1.56E+02 |
| A05 | 113.77 | 22.46 | S-1m | 2.28 | 3.27 | 28.68 | 865.84 | 1.90 | 5.07E+06 ±2.33E+05 | 3.77E+06 ±5.77E+04 | 6.03E+04 ±7.06E+03 | 3.52E+04 ±1.39E+03 |
| | | | B-10m | 14.96 | 2.45 | 26.79 | 1673.87 | 35.10 | 2.04E+07 ±1.92E+05 | 2.93E+07 ±3.61E+05 | 1.92E+04 ±5.36E+02 | 8.13E+01 ±5.26E+00 |
| A01 | 113.65 | 22.74 | S-1m | 0.11 | 2.00 | 28.44 | 2043.89 | 94.78 | 9.76E+06 ±5.80E+05 | 1.74E+06 ±4.56E+05 | 8.79E+04 ±2.43E+03 | 1.92E+04 ±1.42E+03 |
| | | | B-11m | 0.11 | 1.93 | 27.46 | 786.73 | 17.32 | 5.08E+07 ±4.06E+06 | 3.26E+07 ±5.56E+06 | 4.18E+04 ±3.50E+03 | 1.04E+04 ±9.35E+02 |
| F607 | 114.24 | 21.69 | S-1m | 32.74 | 4.88 | 28.74 | 61.84 | ND | 2.08E+03 ±3.57E+02 | 6.07E+04 ±3.75E+03 | 3.70E+01 ±7.50E+00 | 5.30E+02 ±1.88E+02 |
| | | | B-45m | 34.49 | 4.51 | 22.52 | 483.80 | 1.33 | 3.32E+05 ±9.85E+03 | 4.07E+07 ±4.93E+05 | 7.57E+03 ±5.13E+02 | 2.97E+03 ±4.89E+02 |
| F605 | 114.12 | 21.95 | S-1m | 30.11 | 4.64 | 28.10 | ND | 1.91 | 4.98E+03 ±1.16E+03 | 1.29E+06 ±6.16E+04 | 1.11E+02 ±3.14E+01 | 2.07E+03 ±1.56E+02 |
| | | | B-35m | 34.39 | 2.75 | 23.90 | ND | 7.08 | 1.53E+07 ±3.31E+06 | 7.23E+07 ±3.15E+06 | 8.69E+03 ±2.22E+03 | 4.27E+03 ±2.48E+02 |
| F603 | 114.09 | 22.04 | S-1m | 29.09 | 4.46 | 28.30 | 358.38 | 1.68 | 1.78E+03 ±4.75E+02 | 1.44E+06 ±4.94E+05 | 5.56E+01 ±1.38E+01 | 8.82E+02 ±4.80E+01 |
| | | | B-27m | 34.40 | 2.42 | 23.74 | 79.18 | 2.97 | 1.13E+07 ±8.58E+05 | 6.04E+07 ±2.25E+06 | 2.65E+03 ±9.33E+02 | 3.12E+03 ±5.23E+02 |
| F602 | 114.06 | 22.10 | S-1m | 27.08 | 4.86 | 28.96 | ND | 0.33 | 6.10E+03 ±2.52E+03 | 4.69E+05 ±1.54E+05 | 6.18E+01 ±1.19E+01 | 2.17E+02 ±8.47E+01 |
| | | | B-22 | 34.27 | 1.56 | 23.79 | ND | 4.36 | 2.68E+06 ±8.65E+05 | 6.48E+07 ±2.35E+06 | 4.47E+03 ±1.21E+03 | 2.32E+03 ±6.52E+02 |
| F601 | 114.03 | 22.14 | S-1m | 25.32 | 5.09 | 28.38 | 983.39 | 16.09 | 3.58E+04 | 7.92E+04 | 4.85E+01 | 1.29E+03 |

| Station | Lon (E°) | Lat (W°) | Layer | Salinity (PSU) | DO (mg·L⁻¹) | Temperature (°C) | Ammonium (nmol·L⁻¹) | Nitrification rate (nmol·L⁻¹·h⁻¹) | AOA-PA (Copy·L⁻¹) | AOA-FL (Copy·L⁻¹) | AOB-PA (Copy·L⁻¹) | AOB-FL (Copy·L⁻¹) |
|---|---|---|---|---|---|---|---|---|---|---|---|---|
| | | | | | | | | | ±1.26E+03 | ±1.26E+04 | ±2.16E+01 | ±1.18E+02 |
| | | | B-19m | 32.98 | 0.53 | 24.49 | 372.06 | 7.22 | 1.68E+06 | 3.04E+08 | 1.03E+03 | 2.22E+03 |
| | | | | | | | | | ±3.91E+05 | ±4.51E+06 | ±1.03E+02 | ±1.10E+03 |
| | | | S-1m | 26.57 | 4.63 | 28.54 | 1682.83 | 0.51 | 1.33E+03 | 4.86E+05 | ND | ND |
| | | | | | | | | | ±5.22E+02 | ±6.24E+04 | | |
| F701 | 114.18 | 22.14 | | | | | | | | | | |
| | | | B-22m | 34.16 | 1.18 | 23.88 | 1993.45 | 19.13 | 7.90E+05 | 5.41E+07 | ND | ND |
| | | | | | | | | | ±3.50E+04 | ±9.33E+06 | | |
| | | | S-1m | 31.78 | 4.47 | 28.70 | 121.59 | 0.05 | 2.43E+03 | 7.00E+05 | 1.14E+02 | 1.14E+03 |
| | | | | | | | | | ±8.98E+02 | ±1.88E+04 | ±9.51E+01 | ±1.81E+02 |
| F804 | 114.36 | 21.96 | | | | | | | | | | |
| | | | B-29m | 34.47 | 3.46 | 22.91 | 55.20 | 2.86 | 1.47E+07 | 4.71E+07 | 6.91E+03 | 3.16E+03 |
| | | | | | | | | | ±1.69E+06 | ±2.78E+06 | ±3.15E+02 | ±2.24E+03 |

310   * S-Surface; B-Bottom; PA-Particle attached (> 3 μm); FL-Free-living (3-0.2 μm); ND-Under detection limit.

*(Revised) Table S3. Nitrification, community respiration rates and corresponding oxygen demand.*

| Station | Layer | Nitrification rate (nmol·L$^{-1}$·h$^{-1}$) | Nitrification oxygen Demand (mg O$_2$·L$^{-1}$·d$^{-1}$) | Community respiration rate (mg O$_2$·L$^{-1}$·d$^{-1}$) | NOD/CR% |
|---|---|---|---|---|---|
| F101 | S | 1.1770±0.0447 | 0.0014 | 1.4400±0.3024 | 0.094 |
| F101 | B | 36.6152±0.1790 | 0.0422 | 0.1499±0.0021 | 28.137 |
| F104 | S | 0.1443±0.0055 | 0.0002 | 1.6813±0.2433 | 0.010 |
| F104 | B | 0.3277±0.0433 | 0.0004 | 0.1146±0.1568 | 0.330 |
| F107 | S | 0.2057±0.0121 | 0.0002 | 0.2264±0.0722 | 0.105 |
| F107 | B | 0.9596±0.0609 | 0.0011 | 0.2191±0.1756 | 0.505 |
| F301 | S | 5.1961±0.0285 | 0.0060 | 1.1372±0.1240 | 0.526 |
| F301 | B | 41.9434±0.4959 | 0.0483 | 0.4283±0.1175 | 11.282 |
| F303 | S | 0.4847±0.0033 | 0.0006 | 1.0797±0.1843 | 0.052 |
| F303 | B | 36.3678±1.0384 | 0.0419 | 0.5141±0.1635 | 8.150 |
| F305 | S | 1.8411±0.2199 | 0.0021 | 0.6203±0.1090 | 0.342 |
| F305 | B | 1.2795±0.3351 | 0.0015 | 0.0023±0.0017 | 64.894 |
| F701 | S | 0.5144±0.1081 | 0.0006 | 0.9343±0.1157 | 0.063 |
| F701 | B | 19.1291±1.0963 | 0.0220 | 0.0121±0.1519 | 181.913 |
| A14 | S | 0.4443±0.058 | 0.0005 | 1.0191±0.1596 | 0.050 |
| A14 | B | 0.0609±0.0059 | 0.0001 | 0.8222±0.2808 | 0.009 |
| A12 | S | 0.8040±0.0692 | 0.0009 | 0.9928±0.4831 | 0.093 |
| A12 | B | 1.1319±0.0479 | 0.0013 | 0.2256±0.0743 | 0.578 |
| A09 | S | 2.5768±0.1457 | 0.0030 | 1.3144±0.2086 | 0.251 |
| A09 | B | 22.4347±0.6230 | 0.0258 | 0.6340±0.1077 | 4.525 |
| A05 | S | 1.9032±0.186 | 0.0022 | 0.2582±0.0848 | 0.849 |
| A05 | B | 35.0975±2.5993 | 0.0404 | 0.4280±0.0347 | 9.446 |
| A01 | S | 94.7793±12.3754 | 0.1092 | 0.6128±0.1521 | 17.819 |
| A01 | B | 17.3175±0.3106 | 0.0199 | 0.3231±0.1861 | 6.175 |

* S-Surface; B-Bottom.

*(Revised) Table S4. Quantitative PCR results of cDNA (template for RNA level) of AOA and β-AOB in 13 stations*

| Station | AOA-PA (copy·L$^{-1}$) | AOA-FL (copy·L$^{-1}$) | AOB-PA (copy·L$^{-1}$) | AOB-FL (copy·L$^{-1}$) |
|---|---|---|---|---|
| A01 | 3.10E+03 ±1.12E+01 | 3.08E+03 ±7.11E+02 | ND | ND |
| A01 | ND | 1.16E+03 ±7.70E+02 | ND | ND |
| A05 | 8.24E+02 ±4.30E+02 | 1.02E+04 ±1.84E+03 | ND | ND |
| A05 | 1.30E+03 ±8.48E+02 | 6.03E+02 ±3.48E+02 | ND | ND |
| A09 | ND | 1.18E+05 ±1.06E+04 | ND | ND |
| A09 | 1.77E+03 ±1.76E+03 | 1.47E+06 ±1.07E+05 | ND | ND |
| A11 | ND | 2.56E+03 ±8.36E+02 | ND | ND |
| A11 | 3.61E+04 ±3.64E+03 | 1.14E+05 ±1.30E+04 | ND | ND |
| A16 | ND | ND | ND | ND |
| A16 | 2.62E+04 ±6.64E+03 | ND | ND | ND |
| F101 | ND | 1.82E+03 ±5.00E+02 | ND | ND |
| F101 | 7.43E+03 ±1.46E+03 | 1.87E+04 ±2.70E+03 | ND | ND |
| F104 | ND | 1.43E+03 ±4.38E+02 | ND | ND |
| F104 | 1.21E+03 ±7.13E+01 | 8.26E+03 ±8.37E+02 | ND | ND |
| F107 | ND | ND | ND | ND |
| F107 | ND | 1.74E+06 ±5.89E+03 | ND | ND |
| F301 | 2.99E+03 ±1.07E+03 | ND | ND | ND |
| F301 | 5.09E+03 ±1.15E+02 | 1.85E+05 ±1.73E+04 | ND | ND |
| F305 | ND | 8.07E+02 ±5.65E+02 | ND | ND |
| F305 | 1.05E+04 ±1.44E+03 | 9.98E+03 ±1.62E+03 | ND | ND |
| F403 | 6.46E+03 | 1.18E+05 | ND | ND |

| | | | | |
|---|---|---|---|---|
| | ±1.26E+03 | ±1.78E+04 | | |
| F403 | 3.30E+03 | 1.17E+05 | ND | ND |
| | ±1.14E+03 | ±9.54E+03 | | |
| F601 | ND | ND | ND | ND |
| F601 | 4.28E+03 | 3.21E+06 | ND | ND |
| | ±5.20E+02 | ±1.67E+05 | | |
| F603 | ND | 3.72E+03 | ND | ND |
| | | ±3.08E+02 | | |
| F603 | 1.03E+03 | 2.50E+05 | ND | ND |
| | ±7.51E+01 | ±3.04E+04 | | |

* S-Surface; B-Bottom; PA-Particle attached (>3 μm); FL-Free-living (3-0.2 μm); ND-Under detection limit.

---

## Author Comment (AC2) · 14 Aug 2020

Dear Reviewer,

We appreciate your constructive suggestions that have led to an improvement of the manuscript. We have fully addressed these comments during the revision. To assist your assessment of our revised manuscripts, we have provided point-to-point response (**blue in color**) to each of the comments by reviewers below. The location of the change in the revised manuscript is highlighted in our response.

Sincerely yours,

Dr. Hongbin LIU (Corresponding author, Email address: liuhb@ust.hk)

**Response to review 2:**

I feel that this manuscript contains valuable information regarding ammonia oxidizing archaea in estuarine systems, particularly in that it focuses on processes occurring in the water column rather than the sediment, which, as the authors point out, is understudied. However, there are numerous issues with the manuscript in its current form.

*Response: We thank the reviewer for the comments.*

First and foremost, there are serious issues throughout the manuscript with grammar and syntax. Sometimes these issues are so severe that they obscure the meaning of the text. This made it difficult to grasp the authors' meaning and to review the manuscript effectively.

*Response: We thank the reviewer for the comments. We have improved the manuscript by reducing the grammar and syntax as well as following the important suggestions from the reviewer. We have also added detailed information into the method section. We hope that the current version is much clearer.*

In general, the description of the methods is unclear and lacking in detail.

For example: line 78: "the 10-50m by 10m interval" What does this mean?

*Response: We removed "by 10m interval" for the clarity of the station design. The current version is "In the first leg, 83 stations were designed within the 10-50m isobaths covering from the upper estuary to the continental shelf"* Page 4 Line 76-77

lines 87-89: "Sea water was prefiltered... analysis (Liu et al. 2014)." Which analysis was this performed for?

*Response: This sentence described flow cytometry (for microbial cell abundances) sample preparation. For clarity, the current version is "Seawater for microbial abundance quantification was prefiltered by a 20 μm mesh, fixed with final concentration of 0.5 % seawater-buffed paraformaldehyde in cryotubes, and stored in liquid nitrogen until flow cytometric analysis (Liu et al. 2014)."* Page 4 Line 86-88

line 93: "Community respiration rates were measured" in what? Microcosms? Incubations are

mentioned but no volume is given, whether a headspace was left in the bottle...

line 94: "running seawater" Outside the (unmentioned) bottle?

*Response: We have added the corresponding information of community respiration measurement. The running seawater was used to control incubation temperature. The current version is "Community respiration rates (CR) were measured in triplicate in 60ml BOD bottles without headspace through the dissolved oxygen variance before and after 24 h dark incubation submerged in seawater continuously pumped from sea surface"* Page 5 Line 93-111

line 95: "less 10%" Does this mean "less than 10%"?

*Response: Yes. It was revised to "less than 10%".* Page 5 Line 95

line 96: "The del-15N in NO- x the product of nitrification" I have no idea what this means.

line 97: "denitrifier method" What is that? The authors provide citations but for methods but do not explain what they are or how they are performed. Similarly the measurement of the nitrification rate is not described, only cited in an unpublished manuscript.

*Response: We have added the detailed information of nitrification measurement in the revised manuscript. The current version is "Nitrification were measured by incubating $^{15}NH_4^+$ amended (less than 10 % of ambient concentration) seawater in duplicated 200 ml HDPE bottles in dark for 6-12 h, with temperature controlled by running seawater. After incubation, filtrate (0.2 μm-syringe-filtered) was collected and stored in -20 °C for downstream $^{15}NO_x^-$ ($^{15}NO_3^- + {}^{15}NO_2^-$) analysis (Sigman et al. 2001).*

*The nitrification rates were calculated using the following equation:*

$$AO_b = \frac{(R_t NO_x^- \times [NO_x^-]_t) - (R_{t0} NO_x^- \times [NO_x^-]_{t0})}{T} \times \frac{\left[{}^{14}NH_4^+\right] + \left[{}^{15}NH_4^+\right]}{\left[{}^{15}NH_4^+\right]} \qquad (1)$$

*In equation 1, $AO_b$ is the bulk nitrification rate. $R_{t0} NO_x^-$ and $R_t NO_x^-$ are the ratios (%) of $^{15}N$ in the $NO_x^-$ pool measured at the initial ($t_0$) and termination time (t) of the incubation. $[NO_x^-]_{t0}$ and $[NO_x^-]_t$ are the concentration of $NO_x^-$ at the initial and termination of the incubation, respectively. T is the incubation time. $[{}^{14}NH_4^+]$ is the ambient $NH_4^+$ concentration. $[{}^{15}NH_4^+]$ is the final concentration after addition of the stable isotope tracer ($^{15}NH_4^+$). The $NO_x^-$ was*

*completely converted to N₂O by a single strain of denitrifying bacteria (Pseudomonas aureofaciens, ATCC#13985) which lack N₂O-reductase activity (Sigman et al. 2001). The converted N₂O was further analyzed using IRMS (Isotope Ration Mass Spectrometer, Thermo Scientific Delta V Plus) to calculate the isotopic composition of $NO_x^-$. (Sigman et al. 2001; Casciotti et al. 2002; Knapp et al. 2005)."* Page 5 Line 93-111

lines 110-111: "Fast DNA SPIN Kit for Soil" Why would you use a soil kit for filter samples from seawater?

*Response: Our samples spanned from highly turbid riverine water to oceanic waters. For better purification and consistency of our DNA samples, we used the "Fast DNA SPIN kit for Soil". We have used this kit in previous studies, and it works well with plankton samples, so the name of the kit is a bit misleading.*

line 117: "transpired" I assume you mean "transferred"

*Response: Yes. We have revised it into "transferred".* Page 6 Line 123

line 136: "the DNA mixture" I don't know what is meant by this. DNA and cDNA?

*Response: The DNA mixture consisted of 28 DNA samples from 7 stations along A-transect (A01, A05, A09, A11, A12, A14, A16). The DNA mixture here used as a template for clone construction. We want an amoA clone generated from the local community to reduce the dissimilarity between our standard curve and samples.*

Because the methods were so unclear in general, it is difficult for me to assess whether the claims made in the results and discussion sections are to be believed. For example, AOA and AOB copy numbers are referred frequently as evidence of dominance of one group over the other. Is this a rational claim, particularly without 16S data to support it? How many copy numbers of the amoA gene do AOA have vs AOB? And if archaeal amoA transcripts are more abundant than bacterial amoA transcripts, does that mean the archaea are more abundant or simply more active? Is the difference is gene/transcript number statistically significant?

*Response: The amoA gene copies in AOA is one while it is 2-3 copies in AOB (Norton, et al.*

*2002, Hallam et al. 2006). At DNA level, as the amoA gene abundances of AOA in this study were orders of magnitude higher than AOB, we assumed that AOA should be the dominant ammonia oxidizers (Table S2). On the transcript (RNA use cDNA as template) level, we also performed qPCR. We found that AOA were detectable while AOB were under our detection limit (Table S4). Although we cannot rule out the nitrifying activities of AOB by our method, the current evidences supported that AOA is dominant and active in our study.*

As for the measurement of nitrification rates, so little detail is given regarding how these numbers were reached, as to render the data meaningless. The sections on spatial distribution were in general unclear and difficult to follow.

*Response: We have elaborated the nitrification method.* Page 5 Line 93-111

**More specific comments:**

line 223: "B-proteobacteria amoA were under detection limit" Not in all your samples though, judging by Figure 5?

*Response: It is not judged by figure 5. The figure 5 only displayed the size fractionated amoA gene abundance along the A-transect on DNA level. The "under detection limit" is specified for cDNA level in the original sentence. We performed qPCR for both AOA and β-proteobacterial amoA gene abundance using cDNA (represent the RNA level) as template. The data were listed in Supplementary Table S4. Using cDNA as template, we found β-proteobacterial amoA gene abundance were under the detection limit (Table S4).*

line 257: "Besides" Besides what? What is meant by this?

*Response: We have removed "Besides" for clarity.* Page 11 Line 266

line 270: "heterotrophic bacteria abundance" How was this determined? It's not described in the methods.

*Response: We had used the term for all non-phototrophic (no-pigmented) microbial cells in flow cytometric analysis. We admit that flow cytometry method cannot distinguish autotrophic non-phototrophic microbial cells. We have changed "heterotrophic bacteria" into "non-*

*phototrophic prokaryotic cells" with abbreviation "NPC" in the figure legend in Figure 10.*
==Page 33 Line 663-667==; ==Page11 Line 279==; ==Page 15 Line 401-402==

lines 271-272: "Nutrient concentration showed an opposite pattern comparing with salinity" I have no idea what this means.

*Response: We intended to give a general description of the correlation between AOA sublineages and nutrients. Nutrients in PRE were associated with the freshwater discharge. To be clearer, we have revised the sentence as the follow: "In general, WCA sublineages were negatively correlated with nutrient concentration, while SCM1-like sublineages were positively correlated with nutrient concentration."* ==Page 11 Line 280-281==

line 274: "which may be introduced by" Again, no idea.

*Response: We have revised the sentence to "Ammonium showed no significant correlation with AOA sublineages."* ==Page 11 Line280-282==

lines 295-296: "Intensive nitrification... oxygen consumption (Pakulski et al. 1995)." Was that observed in this study or in the study cited?

*Response: It is observed in the cited study. The current version is "Intensive nitrification was observed at intermediate salinities, and it accounted for 20 to over 50 % of oxygen consumption in the Mississippi River plume (Pakulski et al. 1995)"* ==Page 12 Line 302==

lines 300-301: "It is well known... organic matter degradation (respiration)." Be that as it may, you still have to cite it- and it's hardly proof that ammonia is supplied to nitrification by this process.

*Response: We added the citation of paper "Nitrification and ammonification in aquatic systems" (Ward 1996).* ==Page 12 Line 310==

line 305: 229.21% oxygen consumption? How do you consume more than 100% of something in a closed microcosm?

*Response: This may be caused by the methodological difference in the two measurements.*

*Nitrification oxygen consumption were estimated via equation 2 ($NH_3 + 1.5O_2 \rightarrow NO_2^- + H_2O + H^+$). Nitrification in this study are measured in HDPE bottle while community respiration rates were measure in BOD bottles without headspace. We only have one data point at station F701 that exceeding 100%. Similar situation was also observed in Nueces estuary (Yoon and Benner, 1992) and Chang Jiang estuary (Hsiao et al. 2014). Although the unreasonably high NOD/CR ratio might be caused by the underestimated community respiration rates under low oxygen condition (Sampou and Kemp 1994), it showed the potential effect of active nitrification on oxygen consumption in the estuarine system suffered by hypoxia. We have discussed the issue in section 4.1. The oxygen limitation was rather strong for community respiration than nitrification activities (in Section 4.1). Thus, we considered that oxygen consumption via nitrification may contribute to hypoxia formation in the bottom waters.*

lines 328-329: "Though size-fractionated... were observed." I don't understand what is meant here.

*Response: It was a typo, and we mean "Through". We performed qPCR of the size-fractionated (PA-Particle-attached (>3μm) and FL-Free-living (3-0.2μm)) samples. The amoA gene abundances were listed in table S2. Furthermore, figure 5 displayed the amoA gene abundances of the sized-fractionated samples along the A-transect with an increasing salinity gradient. Our result showed differential distribution of the two group of ammonia oxidizers with AOA more abundant in the free-living fraction while AOB more abundant in particle attached fraction and distributed near the upper estuary. We added the citation of figure 5 and Table S2.* Page 13 Line 336-337

line 330: "higher substrate requirement" of what substrate?

*Response: The substate here means "ammonia". We have revised it. Current version is "..higher substrate (ammonia) concentration requirement...".* Page 13 Line 339

In multiple locations in the document the authors mention previous DNA-based studies of AOA and how such studies may overlook active AOA populations. To begin with, those populations would not be overlooked, but perhaps underrepresented in the data. Additionally, several

culture-independent studies of AOA activity utilizing stable isotope probing (in particular, the use of urea as a substrate, and heterotrophy) have been performed in both salt marsh sediment (Seyler et al., 2014, ISME J) and the open ocean (Seyler et al., 2018, FEMS Microbiol Ecol; Seyler et al., 2019, Frontiers Mar Sci), and none of these studies are cited in the text. AOA activity has also been previously described in an estuarine water column using similar techniques to this manuscript (Horak et al., 2013, ISME J; Happel et al., 2018, Env Microbiol)-these should be cited in the text.

*Response: We thank the reviewer for these suggestions. We have revised the statement of "overlooked" or "neglected" into "underrepresented". We have added the citation of Seyler's and Happel's work in the revised manuscript. We have added the citation of Horak's and Happel's work in the revised Table S1.*

*We have cited Seyler's work by adding "Using the stable isotope probing technology, the utilization of organic matter provided evidences of heterotrophy of AOA in the salt marsh sediment and oceanic environment (Seyler, et al. 2014; Seyler et al. 2018; Seyler et al. 2019)." Page 14 Line 395-397.*

*We have cited Happel's work by adding "In Baltic sea, a distinct AOA community were retrieved from RNA level and a few phylotypes related to Nitrosomarinus showed widespread expression in the coastal region (Happel et al. 2018)." Page 13 Line 350-351.*

**As for the figures:**

Figure 6 is impossible to read. Could it be separated into two figures by size fraction? Otherwise there's just too much going on.

*Response: The figure 6 displayed the phylogenetic relationship of top OTUs together with their distinct distribution among samples in the heatmap at both DNA and RNA level. As for the more specific information about the size-fractionated community, we have also displayed in figure 8 by two separated figures. Here, we make a new version for your reference (Figure 6 & 7 below): New Figure 6: Phylogenetic tree and relative abundance (heatmap) of particle attached AOA. New Figure 7: Phylogenetic tree and relative abundance (heatmap) of free-living AOA. Here, we have split original figure 6 into two figures: new figure 6 and new figure 7. The rest of figure legends in the main-text were revised correspondingly.*

[Figure]

*(Revised) Figure 6 Maximum likelihood phylogenetic tree of top 85 OTUs based on amoA gene sequences using T92+G+I model with 1000 bootstrap. The associated heat map is generated based on the relative abundance of top OTUs in the particle-attached samples. Samples are listed from left to right along the ascending salinity gradient.*

[Figure]

*(Newly added) Figure 7. Maximum likelihood phylogenetic tree of top 85 OTUs based on amoA gene sequences using T92+G+I model with 1000 bootstrap. The associated heat map is generated based on the relative abundance of top OTUs in the free-living samples. Samples are listed from left to right along the ascending salinity gradient.*

Figure 7 has me completely puzzled. Firstly because the figure has no axes or scale. Secondly because there's no description of how NDMS analysis was performed in the text. But most importantly, how is it possible that there is absolutely no overlap between the DNA and RNA sequences? I find this incredibly difficult to believe. Are the DNA and RNA sequence data even capturing the same community?

*Response: Figure 7 is NMDS plot generate using Primer 5 (Primer-E-Ltd, PML, UK). The input data was the community composition of 76 samples (OTU table, i.e. relative abundance). The community dissimilarities matrix was calculated using "Bray-Curtis dissimilarity". Thus, the dissimilarity between samples were introduced by compositional difference (different relative abundance of each OTU across all samples). As for the sequence data, for example, the heatmap in figures 6 and 7 has showed the relative abundance of WCA sublinseages*

*presented in both DNA and RNA samples. So, there are shared OTUs in these samples. The archaeal amoA sequencing samples for DNA and RNA (using cDNA as template) were amplified using same primer pair under same conditions and thermal cycles (Francis et al., 2005). The highly dissimilar community composition retrieved from DNA and RNA as well as the differential distribution AOA sublineages is one of our key findings.*

*The previous version generated by Primer 5 cannot show axis information. The current version was generated by R via package "vegan" and "ggplot2" (Oksanen, et al. 2019; Wickham, 2016). The method of NMDS plot has been added into* Page 7 Line 174-177. *This figure is now figure 8 in the revised main text after splitting figure 6 into new figure 6 and new figure 7 according to your suggestion.*

[Figure]

***(Revised) Figure 8. Nonmetric multidimensional scaling (NMDS) plot of AOA community similarity at DNA and RNA level.***

Figure 8 I think is very interesting, but some of the pie charts are so small as to be illegible.

***Response:*** *The revised version is added into the revised manuscript and showed below. The pie charts are enlarged. This figure is now figure 9 after splitting figure 6.*

[Figure]

*(Revised) Figure 9. Free-living and particle-attached AOA community composition and distribution in the Pearl River estuary. The size of the pie charts represents the archaeal amoA gene abundance quantified by qPCR. For a clear display of the AOA community composition, the minimum size of the pie charts is set as 500 copies·L⁻¹. The charts were overlaid on Google Maps (© Google Maps) images using "ggmap" with "ggplot" in R (D. Kahle and H. Wickham, 2013)*

Figure 9 contains some of the most interesting data in the paper, but the figure needs improvement. I think you could combine this heatmap with your phylogenetic tree, and move Figure 6 to supplemental.

*Response: We have followed the suggestions for figure 6 and the figure 9 were replaced with corrected one. Figure 9 is now figure 10 in the revised main text after splitting figure 6.*

| Samples | AOA sublineage | Salinity | NR | DO | $NH_4^+$ | $NO_3^-$ | Tem | $NO_2^-$ | Chl-a | NPC |
|---|---|---|---|---|---|---|---|---|---|---|
| Surface_DNA | WCA I | | | | | | | | | |
| | WCA II | | | | | | | | | |
| | SCM1-like-I | | | | | | | | | |
| | SCM1-like-II | | | | | | | | | |
| | SCM1-like-III | | | | | | | | | |
| | SCM1-like-IV | | | | | | | | | |
| Surface_RNA | WCA I | | | | | | | | | |
| | WCA II | | | | | | | | | |
| | SCM1-like-I | | | | | | | | | |
| | SCM1-like-II | | | | | | | | | |
| | SCM1-like-III | | | | | | | | | |
| | SCM1-like-IV | | | | | | | | | |
| Bottom_DNA | WCA I | | | | | | | | | |
| | WCA II | | | | | | | | | |
| | SCM1-like-I | | | | | | | | | |
| | SCM1-like-II | | | | | | | | | |
| | SCM1-like-III | | | | | | | | | |
| | SCM1-like-IV | | | | | | | | | |
| Bottom_RNA | WCA I | | | | | | | | | |
| | WCA II | | | | | | | | | |
| | SCM1-like-I | | | | | | | | | |
| | SCM1-like-II | | | | | | | | | |
| | SCM1-like-III | | | | | | | | | |
| | SCM1-like-IV | | | | | | | | | |

*(Revised) Figure 10. Spearman correlation between AOA sublineages (relative abundance at DNA and RNA levels) and environmental factors in the surface and bottom layers of the water column in the Pearl River estuary during summer 2017. Only the significant correlations (P<0.05) are displayed (NR-nitrification rates; DO-dissolved oxygen; Tem-Temperature; NPC-non-phototrophic prokaryotic cells).*

Overall I believe the findings presented in this manuscript are likely of interest to the community. The correlations of various AOA lineages to geochemical data and sampling location are very interesting, if difficult to parse in the manuscript's current format. But the issues with the methods in particular and the text in general made it difficult to understand the findings, and some of the claims lack sufficient evidence. I would very much like to see this manuscript again, after significant revisions.

*Response: We thank the reviewer for all insightful and helpful comments. We hope the revised manuscript can meet the standard for publication in Biogeosciences.*

**Reference:**

Casciotti, K. L., D. M. Sigman, M. G. Hastings, J. K. Bohlke, and Hilkert, A. : Measurement of the oxygen isotopic composition of nitrate in seawater and freshwater using the denitrifier method., Anal. Chem., 74, 4905–4912, https://doi.org/10.1021/ac020113w, 2002.

Hallam, S. J. Konstantinidis, K. T. Putnam, N. Schleper, C. Watanabe, Y. Sugahara, J. Preston, C. de la Torre, J. Richardson, P. M. and DeLong, E. F. : Genomic analysis of the uncultivated marine crenarchaeote Cenarchaeum symbiosum, Proc. Natl. Acad. Sci. USA, 103, 48, 18296–18301, https://doi.org/ 10.1073/pnas.0608549103, 2006.

Knapp, A. N., D. M. Sigman, and Lipschultz, F. : N isotopic composition of dissolved organic nitrogen and nitrate at the Bermuda Atlantic time-series study site, Global Biogeochem. Cycle, 19, https://doi.org/10.1029/2004gb002320, 2005.

Norton, J. M. Alzerreca, J. J. Suwa, Y. and Klotz, M. G. : Diversity of ammonia monooxygenase operon in autotrophic ammonia-oxidizing bacteria, Arch. Microbial., 177:139–149 https://doi.org/10.1007/s00203-001-0369-z, 2002.

Oksanen, J., Blanchet, F. G. Friendly, Kindt, M. R. Legendre, P. McGlinn, D. Minchin, P. R. O'Hara, R. B. Simpson, G. L. Solymos, P. M. Stevens, H. H. Szoecs, E. and Wagner, H.: vegan: Community Ecology Package, R package version 2.5-6. https://CRAN.R-project.org/package=vegan, 2019.

Pakulski, J. D., R. Benner, R. Amon, B. Eadie, and Whitledge, T. : Community metabolism and nutrient cycling in the Mississippi River Plume - Evidence for intense nitrification at intermediate salinities, Mar. Ecol. Prog. Ser., 117, 207–218, https://doi.org/10.3354/meps117207, 1995.

Sampou, P. and Kemp, W. N. :Factors regulating phytoplankton community respiration in Chesapeake Bay, Mar. Ecol. Prog. Ser., 110, 249–258, http://doi.org/10.3354/meps110249, 1994.

Sigman, D. M., K. L. Casciotti, M. Andreani, C. Barford, M. Galanter, and Bohlke, J. K. : A bacterial method for the nitrogen isotopic analysis of nitrate in seawater and freshwater, Anal. Chem., 73, 4145–4153, https://doi.org/10.1021/ac010088e, 2001.

Ward, B. B. : Nitrification and ammonification in aquatic systems, Life Support Biosph. Sci., 3, 25–29, 1996.

Wickham, H. : ggplot2: Elegant Graphics for Data Analysis. Springer-Verlag New York, 2016.